



# What determines peat swamp vegetation type in the Central Congo Basin?

Selena Georgiou[1], Edward T.A. Mitchard[1], Bart Crezee[2], Paul I Palmer[1,3], Greta C Dargie[2], Sofie Sjögersten[4], Corneille E.N. Ewango[5,6], Ovide B. Emba[7], Joseph T. Kanyama[5], Pierre Bola[7], Jean-Bosco N. Ndjango[6], Nicholas T. Girkin[8], Yannick E. Bocko[9], Suspense A. Ifo[10], and Simon L Lewis[2,11]

[1]School of GeoSciences, University of Edinburgh, Edinburgh, UK
[2]School of Geography, University of Leeds, Leeds, UK
[3]National Centre of Earth Observation (NCEO), University of Edinburgh, Edinburgh, UK
[4]Faculty of Science, University of Nottingham, Nottingham, UK
[5]Faculté de Gestion des Ressources Naturelles Renouvelables, Université de Kisangani, Kisangani, DRC
[6]Faculté des Sciences, Université de Kisangani, Kisangani, DRC
[7]Institut Supérieur Pédagogique de Mbandaka, Mbandaka, DRC
[8]School of Water, Energy and Environment, Cranfield University, Cranfield, UK
[9]Laboratoire de Botanique et Ecologie, Faculté des Sciences et Techniques, Université Marien Ngouabi, Brazzaville, RoC
[10]10 École Normale Supérieure, Département des Sciences et Vie de la Terre, Laboratoire de Télédétection et d'Écologie Forestière, Université Marien Ngouabi, Brazzaville, RoC
[11]Department of Geography, University College London, London, UK

**Correspondence:** Selena Georgiou (selena.georgiou@ed.ac.uk)

**Abstract.** The Central Congo Basin is home to the largest peat swamp in the tropics. Two major vegetation types overlay the peat: hardwood trees, and palms (mostly the trunkless *Raphia laurentii* variety), with each dominant in different locations. The cause of the location of these differently composed swamp areas is not understood. We investigated their distribution using a recent vegetation classification across the 165,600 km² region. Using a regression model we assessed the impacts of

elevation, seasonal rainfall and temperature on the presence of each peat vegetation type. We used monthly 0.05° resolution CHIRPS rainfall climatology (CHPclim) and maximum temperature (CHIRTS) data together with 90 m resolution terrain data (MERIT Hydro). Our model was successful in predicting the percentage palm swamp composition when tested using data held back for verification, with R² ~ 0.79, RMSE = 14.8%, and p < 0.05 for the largely rain-fed hydrological sub-basins. However, it did not perform well in areas where peatland inundation is controlled by river flooding. We found that palm

swamp composition varies primarily with elevation and dry season climatological variables (rainfall and temperature), with additional, significant contributions from the total wet season rainfall and temperature. There are indications of an optimal range of net water availability (the difference between rainfall and actual evapotranspiration, accounting for run-off) for palm swamp dominance, above and below which hardwood swamp dominates. In this study we progress our understanding of the determinants of peat swamp vegetation type in the central Congo Basin. Improved understanding will contribute to assessing

how changes in environmental factors, including land-use and climate change impacts, could impact swamp type distribution and carbon fluxes in the future.





## 1 Introduction and study area

Peatlands are regions of wetland composed of carbon-rich soil formed from the partial decay of plant materials under water-logged conditions. They are present in over 180 countries (Parish et al., 2008), with tropical peatlands existing in Southeast Asia, Africa, Central America, South America and the Caribbean (Page et al., 2011). Xu et al. (2018) conducted a meta-analysis of research assessing the spatial extent of peatlands and estimated global peatland area to be 4.23 million km$^2$, covering ~2.84% of land globally, with 187,061 km$^2$ located in Africa. The total estimated global peatland carbon stock is 600 to 650 Gigatonnes of carbon (Gt C) (Yu et al., 2010; Loisel et al., 2021; Page et al., 2022), of which tropical peatlands have been estimated to contribute between 10 and 30% (Yu et al., 2010; Page et al., 2011, 2022; Dargie et al., 2017; Hodgkins et al., 2018). The characteristics of tropical peatlands in Southeast Asia are comparatively well understood, with their extent (Page et al., 2007), depth and biomass attributes quantified (Hooijer et al., 2010; Page et al., 2011). However, peatlands in other tropical regions, including the Cuvette Centrale, have, until recently, remained largely unexplored. They form major stores of carbon, having accumulated peat and acted as a carbon store for at least 10,000 years (Page and Baird, 2016; Dargie et al., 2017), and are now vulnerable to rapid loss of their carbon stocks through land-use change and climate change impacts (Dargie et al., 2019).

The Cuvette Centrale (Figure 1), situated in the central lowland region of the Congo Basin, is the second largest wetland in the tropics (Dargie et al., 2017) and contains the largest area of tropical peatland, with an estimated extent of 165,600 km$^2$, and storing 29.0 Pg C (95% CI, 26.3-32.2 Pg C) (Crezee et al., 2022). It spans the Republic of Congo (RoC) and the Democratic Republic of Congo (DRC), with the Ubangi river marking the border between these two countries.

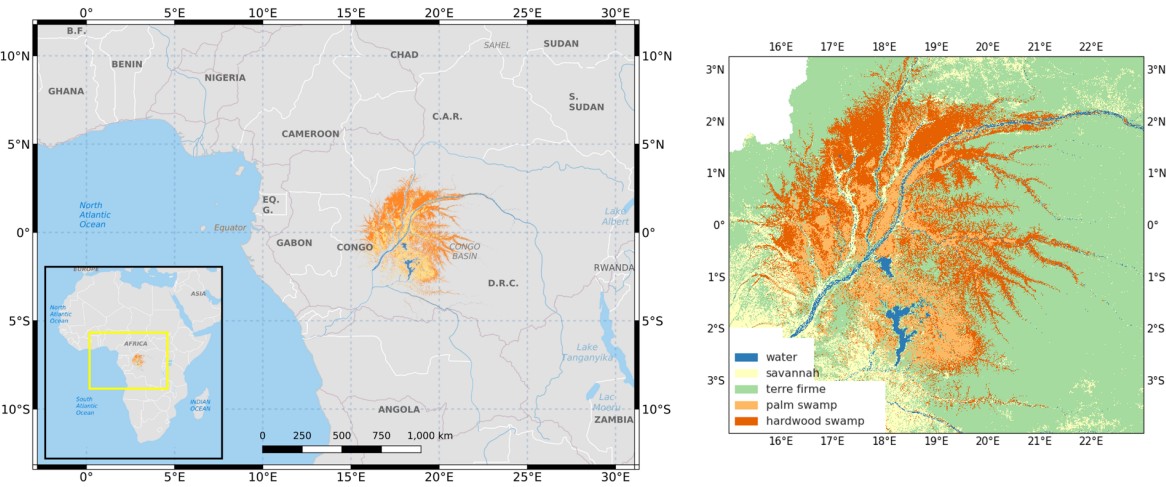

**Figure 1.** Location of the Cuvette Centrale peatland region within the central Congo Basin. Light orange represents regions of palm swamp (largely *Raphia laurentii*), and dark orange represents regions of hardwood swamp trees. The map on the right shows the full land type classification map for the Cuvette Centrale, developed by Crezee et al. (2022).

Dargie et al. (2017) estimated that the Cuvette Centrale's carbon storage contribution increases the global carbon stock within tropical peatlands by 36% to 104.7 Pg C (within a 69.6 to 129.8 Pg C range). They identified the two main types of





swamp vegetation which overlay the region's peat as: hardwood trees, including the *Uapaca paludosa*, *Carapa procera* and *Xylopia rubescens* species; and palms (largely *Raphia laurentii*). They also calculated mean above ground carbon stocks to be 84% higher for hardwood than for palm swamp plots. However, they found no evidence of differences in carbon concentration, peat depth or bulk density, and, therefore, below-ground carbon stocks, between the two swamp types. Crezee et al. (2022)

modelled peat depth and carbon density values across the peat swamp and found these values to be slightly higher for palm swamp (1.7 (±0.8) m for hardwood, versus 1.9 (±0.9) m for palm swamp). The mean carbon density is 1725 (±617) Mg/ha and 1826 (±674) Mg/ha over hardwood and palm swamps, respectively. An estimated area of 56,300 (±5,600) km² palm dominated swamp across the Cuvette Centrale was derived from the Crezee et al. (2022) land classification map, indicating that *Raphia laurentii* could be the most common palm in Africa. However, we do not know why it is distributed where it is.

Understanding the determinants of peat swamp vegetation type is important, as changes in environmental factors, including climate change impacts, could lead to preferential conditions for one type over another, which, in turn, could impact soil carbon fluxes (Loisel et al., 2021). Gutenberg et al. (2019) determined that soil carbon flux depends significantly on forest type, as well as soil moisture and temperature, when investigating a freshwater forested peatland in the USA, and Sjögersten et al. (2018) showed that greenhouse gas (GHG) emissions from tropical peatlands in Panama vary by vegetation type, with net

emissions of both carbon dioxide ($CO_2$) and methane ($CH_4$) observed to be highest from palm swamp forest under all levels of peat moisture investigated. They also identified the greatest increases in $CH_4$ emissions with temperature for palm swamp forest. It is therefore important to understand how climate change may impact the distribution of peat swamp vegetation type, as well as to consider the response of different peatland swamp vegetation types to future climate change when predicting GHG emissions.

Our understanding of the hydrological processes that take place across the peatlands is still developing (Biddulph et al., 2021). The Cuvette Centrale is fed by many tributaries, with the inter-fluvial basins differing in soil characteristics and lithology, in addition to rainfall distribution (Borges et al., 2019). Peatland swamp areas within the Cuvette Centrale are waterlogged for much, or all, of the year, enabling the gradual accumulation of peat. Some regions are river-influenced, receiving water input from the river system in addition to rainfall, while the large inter-fluvial basins are believed to be mainly rain-fed

(ombrotrophic), due to being elevated from the river system, and cut-off from surface or sub-surface hydrological input and flood waves from the water channels (Dargie et al., 2017). Satellite data based studies by Jung et al. (2010) and Lee et al. (2011) support that the Congo Basin wetlands are largely independent from the river system. Typically, rain-fed peatlands have a dome structure (where the peat is thickest at the central point, and thins towards the margins) (Lähteenoja and Page, 2011), and although pronounced dome structures are not evident in the Cuvette Centrale, recent work (Davenport et al., 2020) has

confirmed shallow dome structure (1.8 m over ~20 km), at a single site in the RoC.

The peat swamp regions of the Cuvette Centrale typically receive 1700 mm rainfall annually (Samba et al., 2008), significantly less than other regions where tropical peatland exists, for example, the Pastaza Marañón Foreland Basin, Peru, which receives around 3000 mm/year (Marengo, 1998). Seasonal variations in rainfall and temperature across the Cuvette Centrale are driven by the movement of the inter-tropical convergence zone (ITCZ), which is a band of low pressure that moves with the

thermal equator, bringing increased rainfall to the Cuvette Centrale during two wet season periods: March, April, May (MAM),




as it moves northwards from the equator, and September, October, November (SON), as it moves back southwards bringing the heaviest period of seasonal rainfall. Two dry season periods are observed in-between: December, January, February (DJF) and June, July, August (JJA). As a result, the water table level across most of the peatlands can have a strong seasonality. Hirata et al. (2015) describe hydrology as a key component of ecosystem models, especially variations in the water table level which affect rates of both photosynthesis and plant decomposition (Hirano et al., 2007, 2012; Jans et al., 2012). It is not currently understood how swamp vegetation type varies under different levels of rainfall and flooding in the Cuvette Centrale, and it is possible that climate change could result in ecosystem adaptation to a different distribution of peatland vegetation.

Within this study our aim was to improve our understanding of climatic influences on the current distribution of Cuvette Centrale peatland vegetation. The peatland maps produced by Dargie et al. (2017) and Crezee et al. (2022) show that palm-dominated swamp forest tends to occur in more interior peatland locations, with hardwood-dominated more towards the edge, delineated by the river system. We hypothesise that: 1. additional water input in the form of higher water tables and/or more extensive periods of water availability benefits palms, and that areas where palm dominates benefit from higher dry and/or wet season rainfall totals, and 2. palms may dominate at lower elevations to benefit from additional water input from ground water flow and pooling of water.

Currently, GHG fluxes from peatlands are not incorporated within Earth System Models (ESMs) (Loisel et al., 2021). An improved understanding of these relationships would contribute usefully to informed integration of tropical peatland dynamics into future ESM implementations, such that we can better simulate how future land-use and climatic change will impact ecosystem dynamics and GHG fluxes. To assess these hypotheses we perform a regression analysis to quantify the impact of elevation and seasonal climatological variables on sub-basin swamp vegetation type.

## 2  Data

We use the  50 m land-type classification map (figure 1) developed by Crezee et al. (2022), together with terrain and climatological data to assess the drivers of differences in regional composition between the two major swamp vegetation types, palm and hardwood. To summarise, the terrain, climatological and weather data we use includes: the MERIT Hydro 90 m elevation and Height Above Nearest Drainage (HAND) basin (Yamazaki et al., 2017); Hydrobasins sub-basin delineations (Lehner and Grill, 2013); Climate Hazards center Infrared Precipitation with Stations (CHIRPS) rainfall (Funk et al., 2015a); CHPclim monthly rainfall climatology (Funk et al., 2015b); and CHIRTS maximum temperature (Funk et al., 2019). This section provides a more detailed description of these data.

### 2.1  Topographical data

Although the peatland swamp area of the Cuvette Centrale is relatively flat with most areas lying between 300 and 340 m above sea level (a.s.l), there is sufficient variation in the surrounding topography (up to 800 m a.s.l) for surface and sub-surface run-off to occur, potentially carrying rain water inputs from higher elevations to the lowland swamp regions. It is necessary for Digital Elevation Models (DEMs) to be as accurate as possible when assessing hydrological and carbon cycle dynamics



(Yamazaki et al., 2017), however, wetland regions of the world, including the Congo Basin, are significantly affected by tree height bias in DEMs. To assess if run-off contributes to net-water input at lower elevations, we use the MERIT hydrologically adjusted DEM, available at ~90 m resolution (Figure 2a). It was developed using the NASA SRTM3 and the JAXA AW3D 30 m DEM as baseline products, with bias, noise and tree-height corrections applied to approximate the terrain elevation, and with the Congo Basin being included as one of the focal regions when developing the product (Yamazaki et al., 2017).

Nobre (2011) introduced the Height Above Nearest Drainage (HAND) basin model which calculates the terrain elevation relative to the local drainage network. We make use of the MERIT derived HAND data (Yamazaki et al., 2017), averaged over 0.05° latitude x 0.1° longitude sub-regions, which equates to cell areas of ~62 km$^2$ or 6200 hectares (ha).

Additionally, we use the HydroBASINS sub-basin delineations. This is a global collection of shape files at 15 arc second resolution (~450 m at the equator), derived from the HydroSHEDS database (Lehner and Grill, 2013), which details basin boundaries and sub-basin delineations at successive levels of detail (Figure 2b).

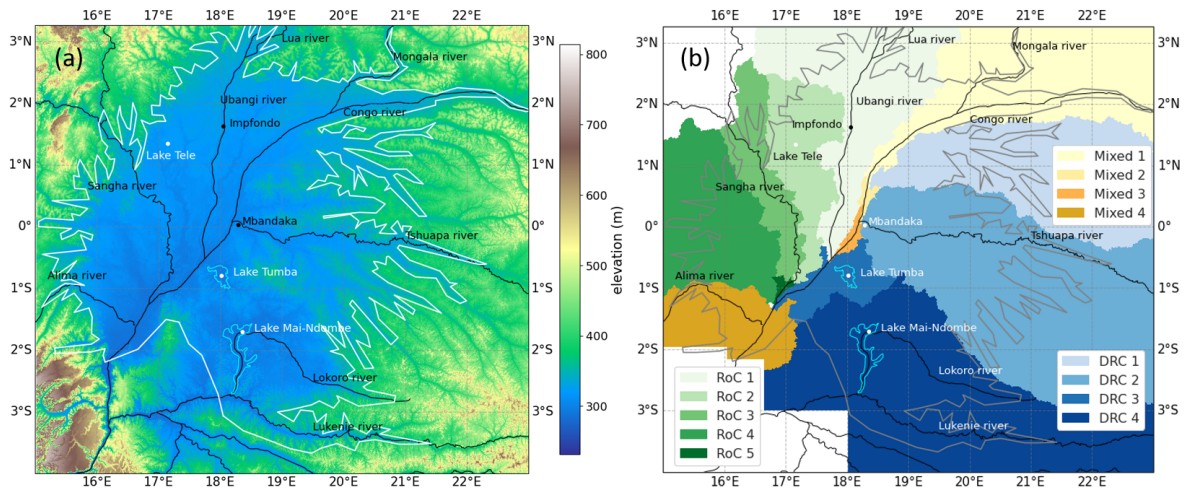

**Figure 2.** (a) MERIT elevation data, and (b) Sub-basin map, derived using Hydro-basins level 4 and 5 sub-basin shapefiles. The major rivers, towns, and lakes are annotated, and the approximate footprint of the peatland complex is overlaid.

## 2.2 Climatological and meteorological data

The CHIRPS rainfall dataset is available on a daily basis at 0.05° resolution (5.55 km resolution at the equator) between approximately 40° N and 40° S. It has been derived from a combination of remotely sensed rainfall and gauge data (Funk et al., 2014). Camberlin et al. (2019) performed an inter-comparison study of remotely sensed rainfall products available over Central Africa, including five data sets incorporating ground-based rain gauge measurements, of which CHIRPS has the highest spatial resolution. The Tropical Rainfall Measuring Mission (TRMM) Multi-satellite Precipitation Analysis product was found to have the smallest bias over Central Africa when compared with other rainfall products, however its spatial resolution is 0.25° (~28 km at the equator). They found that all products reproduced the spatial variability of the rainfall well, and CHIRPS



and TRMM performed favourably at the inter-annual scale. Due to the high seasonal variability of rainfall within the Cuvette Centrale, historical rainfall data of sufficiently high resolution was required for this study, so we decided to use the CHIRPS data. Within this investigation, we make use of both the CHPclim (1980 to 2009) climatology data (Funk et al., 2015b) and

125 the monthly CHIRPS data, available on a yearly basis from 1981 to present (Funk et al., 2014, 2015a). Figures 3a and b show the annual and seasonal rainfall climatology spatial distribution. Additionally, we use the Climate Hazards Center (CHC) CHIRTS monthly maximum temperature data (Funk et al., 2019), also available at 0.05° resolution, to calculate mean seasonal climatologies within the time-frame over which we calculated the CHIRPS rainfall climatology (Figures 3c and d). The spatial pattern of evapotranspiration does not vary greatly across the Congo Basin (Bultot and Griffiths, 1972; Alsdorf et al., 2016).

We therefore effectively use rainfall totals as a proxy for the pattern of net water availability in rain-fed regions of the Cuvette Centrale. CHIRTS temperature data is only available from 1983. We use the 27 year period, 1983 to 2009, to overlap with the 1980 to 2009 CHIRPS rainfall climatology. Although there is a three year difference in the averaging periods, we use them over a sufficiently long period and concurrent time-frame for their values to be representative with respect to one another.

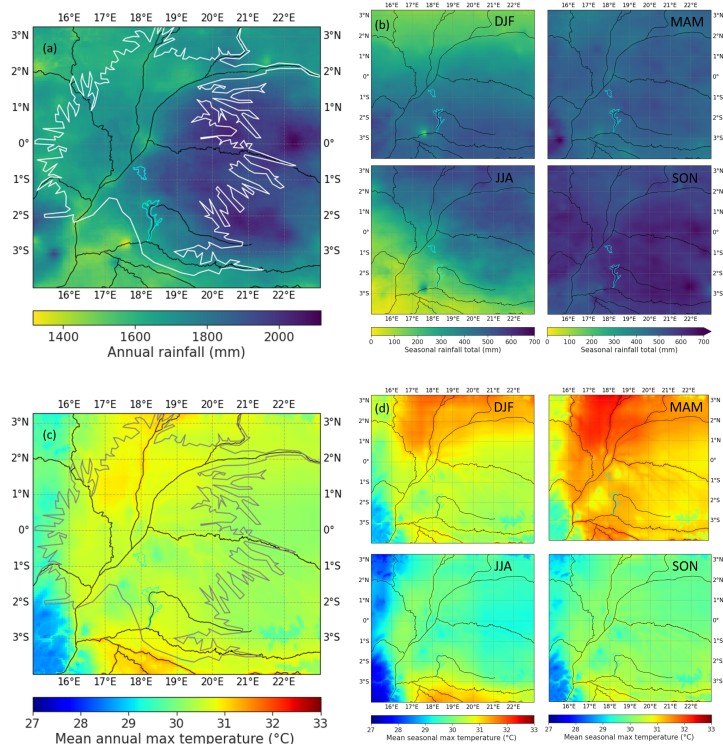

**Figure 3.** (a) Multi-year mean (1980 to 2009) annual rainfall accumulation and (b) seasonal rainfall accumulation climatologies for the dry season periods: Dec, Jan and Feb (DJF) and Jun, Jul, Aug (JJA) and the wet periods: Mar, Apr, May (MAM) and Sept, Oct, Nov (SON). Derived from CHPClim data (Funk et al., 2015b). (c) Multi-year mean (1983 to 2009) climatology of the monthly maximum temperature CHIRTS data, and (d) corresponding seasonal means of the monthly maximum temperature climatology. Black lines denote the main river tributaries, blue lines show lakes, and the white and grey borders in (a) and (c) indicate the approximate footprint of the peatland complex.





## 3  Methods

### 3.1  Sub-basin delineation

We assess the relationships between the climatological data and vegetation type at the sub-basin scale due to regional meteorological differences. We use the level four and five HydroBASINS layers to sub-divide the Cuvette Centrale into 13 sub-basins (Figure 2b). We further categorise the sub-basins into three regions based on their relative location to the Congo river:

    i.  RoC: sub-basins lie within the RoC (RoC 2 to 5) or between the RoC and DRC, on the right bank of the Congo river, stretching across the Ubangi river (RoC 1).

    ii.  Mixed: sub-basins span across the left and right banks of the Congo river (Mixed 1 to 4).

    iii.  DRC: sub-basins lie fully, or largely, in the DRC, and on the left-bank of the Congo river (DRC 1 to 4).

### 3.2  Derivation of seasonal climatological data

We use the CHPclim and CHIRTS data to calculate three-month seasonal climatologies for the Cuvette Centrale across the dry seasons (December to February (DJF) and June to August (JJA)) and wet seasons (March to May (MAM) and September to November (SON)). Additionally, we derive climatological variables from the four three-month seasonal periods, including: dry and wet season rainfall totals (DJF + JJA, and MAM + SON); dry and wet season differences in rainfall (JJA-DJF and SON-MAM); and the difference in mean maximum temperature between subsequent seasons. We assess the spatial correlations of the derived seasonal rainfall and maximum temperature climatologies with the swamp vegetation type at the sub-basin level.

### 3.3  Implementation of a fine grid method to assess drivers of swamp vegetation type

Due to the high variability in rainfall totals across the Cuvette Centrale, indicated in Figure 3a, we further divide the region into 0.05° latitudinal x 0.1° longitudinal pixels (approximately 5.55 x 11.1 km resolution/6200 ha cells). We use different resolutions for each sub-pixel dimension as, although the rainfall pattern varies significantly across the basin, there is less seasonal variability in the longitudinal direction than the latitudinal due to the north-southwards migration of the ITCZ (Figure 3b).

Our analysis involves calculating first the percentage composition of each swamp vegetation type within each sub-pixel, and then the palm and hardwood swamp composition as percentages of the total swamp composition (hardwood + palm swamp pixels). To limit the regression analysis to understanding the impact of climatological drivers on regions of swamp vegetation, we include only sub-pixels with greater than 70% total swamp composition (hardwood and palm swamp combined).

### 3.4  Selection of feature variables to include in the regression model

We calculate the Pearson correlations between all our derived topographical and climatological variables, and the percentage palm swamp composition to identify where high collinearity exists. Ideally, only feature variables that are sufficiently independent from one another should be included in regression analyses, such that the output parameter coefficients are representative




of the relative contribution each parameter makes to the prediction of palm swamp composition. The inclusion of too many multi-collinear variables in the regression method we use also results in non-convergence of the model. We therefore use the Variation Inflation Factor (VIF) method to calculate multi-collinearity between different combinations of feature variables, and to arrive at a final set of suitable feature variables for inclusion in our regression model implementation.

As we average the MERIT 90 m elevation dataset over ~6200 ha areas, the corresponding standard deviations will be large for pixels with large sub-grid variations in elevation. Where this is the case, there may be additional surface/sub-surface run-off between sub-pixels to consider, and our linear regression model would be less representative as rainfall input at these locations would not necessarily correspond with total net water input without further accounting for hydrological mechanisms due to the terrain. We therefore include elevation standard deviation as an additional feature variable within the model.

We calculate linear regressions between palm swamp composition and the elevation, elevation standard deviation, rainfall and temperature variables, for each of the three regions: RoC, DRC and Mixed, and for each sub-basin within these regions. These regional and sub-basin distinctions are made to (i) assess overall inter-regional differences resulting from differing hydrological mechanisms (e.g. river flooding in the Mixed and DRC regions, and higher rainfall totals over the DRC peatland regions), and (ii) to assess intra-regional differences. Due to differences in hydrological regimes across the Cuvette Centrale, and the influence of additional water inputs in the floodplain regions, we limit our regression analysis to the sub-basins believed to be mostly rain-fed (RoC and Mixed sub-basins), such that the seasonal rainfall patterns can be regarded as being largely representative of the spatio-temporal patterns of net water input for a particular region, enabling us to delineate the impact of different levels of rainfall on swamp vegetation type dominance.

In addition to our final choice of model variables, we trial two other models, one looking at how well the seasonal rainfall parameters alone can be used to model palm swamp composition, and another using seasonal maximum temperature parameters as an alternative.

## 3.5 Regression model implementation

The variables used within regression models should ideally be independent, however, due to the impacts of multi-collinearity, our choice of some variables effectively act as interaction terms, influencing the size and significance of the model's determined coefficients through their influence on one another's contribution to predicting the dependent variable, the percentage palm swamp composition. Multicollinearity only impacts interpreting the modelled significance of variables which are multicollinear, and not that of other independent variables. Additionally, although correlation between model features can affect the coefficients and p-values, it does not mean that a good model fit cannot be found, and useful predictions made (Neter et al., 1997).

Douma and Weedon (2019) provide an overview of statistical regression methods for modelling proportional data, distinguishing between count-based and continuous proportions. Count-based proportions can be modelled well using logistic regression, while they identify Beta and Dirichlet regression methods as being appropriate for continuous proportions, e.g. percentage cover, which is what we aim to do within this study when modelling for percentage palm swamp composition. Additionally, logistic regression methods are suitable for continuous proportions, where the distribution is Gaussian. How-





ever, due to the distribution of our dependent variable being left-skewed and non-Gaussian, we use a Beta-regression method with a logistic link function. We use the R code betareg implementation of the beta regression method, developed by Grün et al. (2012), and described by Cribari-Neto and Zeileis (2010). It uses maximum likelihood estimation (MLE) to estimate the

parameters of the probability distribution, given the independent data inputs. It is similar to a generalized linear model, and can be used to predict non-Gaussian proportional data, which was required in our use-case. In addition to dealing well with asymmetric and skewed data, an advantage of using a beta regression model over a linear model with logistic input is that the contribution of each variable to the prediction can be determined from the size of the multiplying (beta) coefficients when using standardised feature variable inputs.

We standardise the independent variables using the Z-score method (see equation A1 in Appendix A), which expresses the variable in terms of the number of standard deviations its value lies from the population mean value. We then use these standardised values as inputs to the model, such that the derived model linear regression coefficients represent the relative importance of each feature variable in influencing our dependent variable, the palm swamp composition of a particular 0.05° x 0.1° pixel. We then divide our data from the selected sub-basins (981 points), where total swamp composition is greater than

70%, into 80% training (785 points) and 20% test (196 points) data sets using the train-test split function available within the Scikit-learn python package. It is important to split the data such that, by comparison of the output statistics from the model predictions for each data set, we can ensure that our model does not over-fit to the training data. We create 10 different train-test data split combinations, by defining different random-seed numbers within the train-test split function, and run the model for each dataset to assess the stability of the $\beta$ coefficients output from the model. The pseudo-$R^2$ value differs from the more

commonly used $R^2$ statistic, and is calculated within the R Betareg package as the squared sample correlation between the linear predictor and the logistically transformed response (Ferrari and Cribari-Neto, 2004; Cribari-Neto and Zeileis, 2010).

### 3.6 Assessing the model anomalies

To assess where outliers exist, we plot the differences between the mapped and predicted values across the full dataset (981 6200 ha cells). We define outliers as being located outwith +/- two times the standard deviation ($\sigma$). To understand the reasons

behind the anomalies, we investigate how they vary with both the feature variables included in the regression model, and also with those not included due to multi-correlation. We calculate the Pearson correlation between the anomalies and each variable of interest to assess if there are any relationships not well accounted for within our model.

### 3.7 Assessing the contribution of inter-annual variability in seasonal climatology

The El Niño-Southern Oscillation (ENSO), describes irregular changes in pressure between the East and West regions of

the Tropical Pacific, and the resulting impacts on sea surface temperature (SST) and the Walker Circulation (an atmospheric circulation which is driven by the equatorial SST gradient across the Pacific Ocean) which can culminate in El Niño and La Niña episodes (Oliver, 2005). Amarasekera et al. (1997) identified a weak negative correlation between annual discharge from the Congo Basin and the equatorial Pacific SST anomalies associated with ENSO. They estimate that 10% of variance in the Congo Basin's annual discharge can be attributed to ENSO.



So far, the methodology we have described aims to assess the contribution of climatological conditions to determining the prevalence of each swamp vegetation type. We are also interested in understanding if the inter-annual variability, which is driven to some degree by ENSO conditions, can serve to delineate the preferential conditions for each swamp type to dominate, and explain the presence of any outliers calculated using our beta regression model implementation. To assess this we use the monthly CHIRPS rainfall and CHIRTS maximum temperature data available since 1981 and 1983 to present, respectively. We

calculate the maximum, minimum and standard deviation of each dataset on a pixel by pixel basis corresponding with each of the climatological variables used within our original model implementation. We then identify subsets of years that were drier and wetter than the mean values over the 1981 to 2010 period on an individual sub-basin basis, and calculate climatologies over each of these sets of years. We then re-run our model using these dry/wet climatologies to find if inter-annual variations in rainfall accumulations impact on swamp type composition. Additionally, we compare yearly seasonal rainfall totals with the

corresponding ENSO index for that season, using the ENSO indexes and durations available on the NOAA ENSO monitoring website: https://origin.cpc.ncep.noaa.gov/products/analysis_monitoring/ensostuff/ONI_v5.php.

## 4   Results

Our model implementation successfully predicted palm swamp composition over the 6200 ha sub-regions (Figure 4) included within the 20% test data set we had held back for verification purposes ($R^2$=0.79, RMSE = 14.8%, $p < 0.05$), and demonstrates

the strong dependence of swamp vegetation type on elevation and climatological variables, primarily the dry season rainfall and temperature (Table 1). We provide here an overview of the results from the correlation and regression analyses, including a summary of how our model implementation performed over each sub-basin, and taking into consideration outlying predictions.

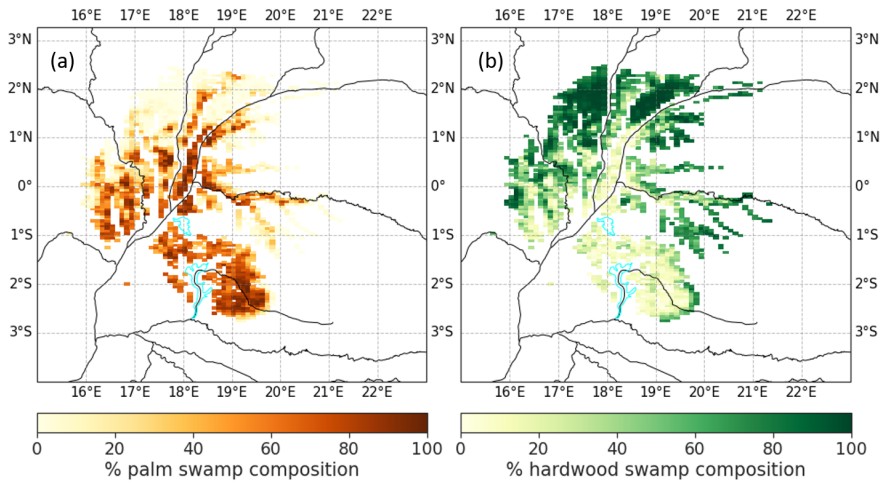

**Figure 4.** Maps show the spatial distribution of regions containing greater than 70% total swamp composition (palm + hardwood) within each 0.05° by 0.1° (6200 ha) sub-pixel, expressed as percentage composition of (a) palm swamp, and (b) hardwood swamp. Only pixels meeting this criteria were used within the regression analysis. The major lakes (blue) and rivers (black) are shown.





**Table 1.** Summary statistics for each of the three sets of feature variables used within the beta regression model. The median residual value is weighted and standardised.

| Model name and variables | coefficient | P-values | Pseudo-$R^2$ value | Log-likelihood | median residual |
|---|---|---|---|---|---|
| **1. Elevation + rain + temp** | | **all < 0.05** | 0.75 | 939.5 | -0.0616 |
|    i. Elevation | -4.64 | < 2e-16 | | | |
|    ii. Elevation standard dev | -0.30 | 0.03457 | | | |
|    iii. JJA-DJF rainfall | 3.23 | < 2e-16 | | | |
|    iv. DJF total rainfall | 0.67 | 0.01393 | | | |
|    v. Total wet season rainfall | 0.73 | < 2e-16 | | | |
|    vi. mean DJF Tmax | -1.28 | < 2e-16 | | | |
|    vii. mean SON Tmax | -0.35 | 0.00153 | | | |
| **2. Rainfall only** | | **all < 0.05** | 0.50 | 698 | 0.0221 |
|    i. JJA-DJF rainfall | 2.59 | < 2e-16 | | | |
|    ii. DJF total rainfall | 4.28 | < 2e-16 | | | |
|    iii. Total wet season rainfall | -0.18 | 0.0139 | | | |
| **3. Temperature only** | | **all < 0.05** | 0.39 | 606.5 | 0.0303 |
|    i. mean SON Tmax | -0.69 | < 2e-16 | | | |
|    ii. mean DJF Tmax | 1.07 | < 2e-16 | | | |

## 4.1 Correlation analyses

### 4.1.1 The impact of elevation on swamp vegetation type

We observe negative correlations between elevation and palm swamp composition, and that palm swamp dominates at the lowest elevations across all three regions (Figures 5a-c). Although the slopes of these linear relationships are similar, the intercepts vary between sub-basins for the RoC and Mixed regions. This indicates that the dependence of swamp type composition on elevation is relative to the local terrain within each sub-basin, rather than dependent on larger scale considerations, for example the change in temperature with elevation.





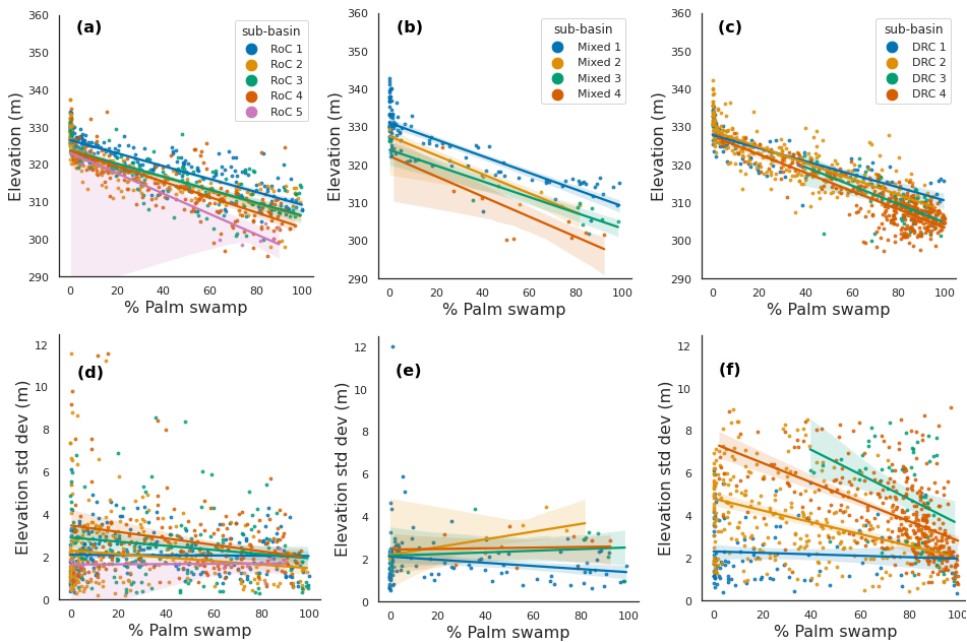

**Figure 5.** Variations in palm swamp composition with: elevation for (a) RoC (b) Mixed and (c) DRC sub-basins, and with the standard deviation of the elevation for (d) RoC (e) Mixed and (f) DRC sub-basins. 0% palm swamp is equivalent to 100% hardwood swamp and vice versa. The shaded envelopes represent the standard deviation. Some of the linear regressions over smaller sub-basins included less than 10 pixels, resulting in higher uncertainty.

We observe a slight negative correlation between palm swamp composition and elevation standard deviation across most RoC sub-basins (Figure 5d). These linear regressions have significant (p<0.05) relationships for RoC sub-basins 2 to 5, where groundwater flow may contribute more to total net water availability. The relationship between palm swamp composition and elevation standard deviation is far more significant in the DRC for sub-basins DRC 2 to 4 (Figure 5f), indicating a higher likelihood of additional net water contribution from ground water flow in these regions.

### 4.1.2 Annual rainfall correlations with swamp type dominance

There are significant positive correlations between palm swamp composition and the annual rainfall accumulation for all RoC sub-basins (Figure 6a). We observe a similar trend for the Mixed 1 sub-basin which lies wholly in the DRC, but with most of its peatland area located between the Ubangi and Congo rivers, and also for the Mixed 2 sub-basin which is spread across the Congo river (Figure 6b). However, comparing the linear regressions for the Mixed 1 sub-basin (north of the Congo river) and its neighbouring DRC 1 sub-basin (just south of the Congo river) we observe opposing trends (Figure 3a and b), with Mixed 1 showing more similarity to the RoC sub-basins in terms of both rainfall totals and slope. The Mixed 1 sub-basin region is extensive, but the peatlands located within it are largely located next to the Congo mainstem, and are therefore susceptible to





receiving additional water input from river flooding. Contrastingly, palm swamp composition has a negative correlation with annual rainfall totals for DRC sub-basins 1, 2 and 4.

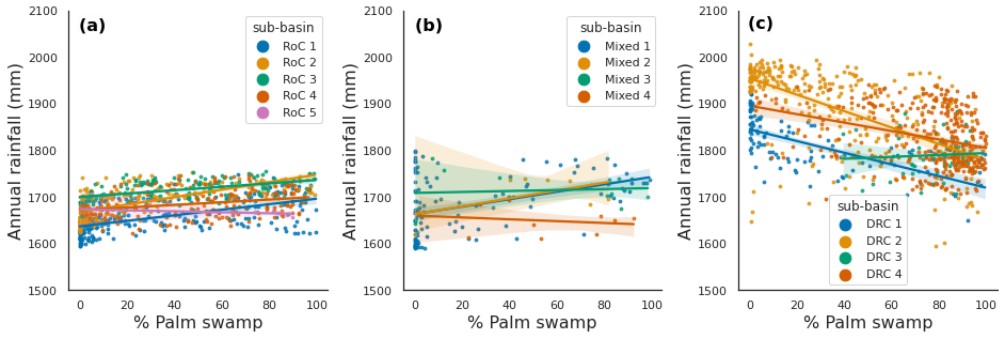

**Figure 6.** Linear regressions between palm swamp composition and annual rainfall totals for (a) RoC (b) Mixed and (c) DRC sub-basins. 0% palm swamp is equivalent to 100% hardwood swamp and vice versa.

### 4.1.3 Seasonal rainfall correlations with palm swamp composition

By comparison with the annual rainfall correlations in Figure 6 we observe that the spread in annual rainfall for the RoC and Mixed sub-basins is largely accounted for within the spread of the DJF rainfall totals (Figures 7a and b), indicating that variation in palm swamp dominance due to rainfall can be mainly attributed to spatial differences in dry season rainfall. Contrastingly, for most of the DRC sub-basins, the negative correlation between palm swamp composition and annual rainfall accumulations is weakly attributable to wet season rainfall totals. This supports that the climatological relationships that impact on swamp type composition within the RoC and Mixed sub-basins differ substantially from those for the DRC sub-basins.

In addition to the individual dry season rainfall accumulations, the difference between the first and second dry season rainfall accumulations was found to have a significant relationship with palm swamp composition across the RoC and Mixed sub-basins (Figures 7d-f). We observe that the relative difference in rainfall between these two seasons is of more significance than the actual rainfall difference for each sub-basin, with each sub-basin's linear regression having a different intercept value. There is less variation in the second dry season rainfall (JJA) than in the first (DJF), and therefore, the higher the DJF rainfall, the smaller the dry season difference, and this corresponds with increasing likelihood of palm swamp dominance.

Overall, palm swamp composition was found to have more significant correlations (higher $R^2$ values) with the total wet season rainfall accumulation (MAM + SON) than with the annual rainfall total (Figures 7g-i). As with the annual rainfall (Figure 6) we observe significant positive correlations between palm swamp composition and the total wet season rainfall for RoC sub-basins, and significant negative correlations for DRC basins, with the exception of DRC 3 for which there is no significant relationship.



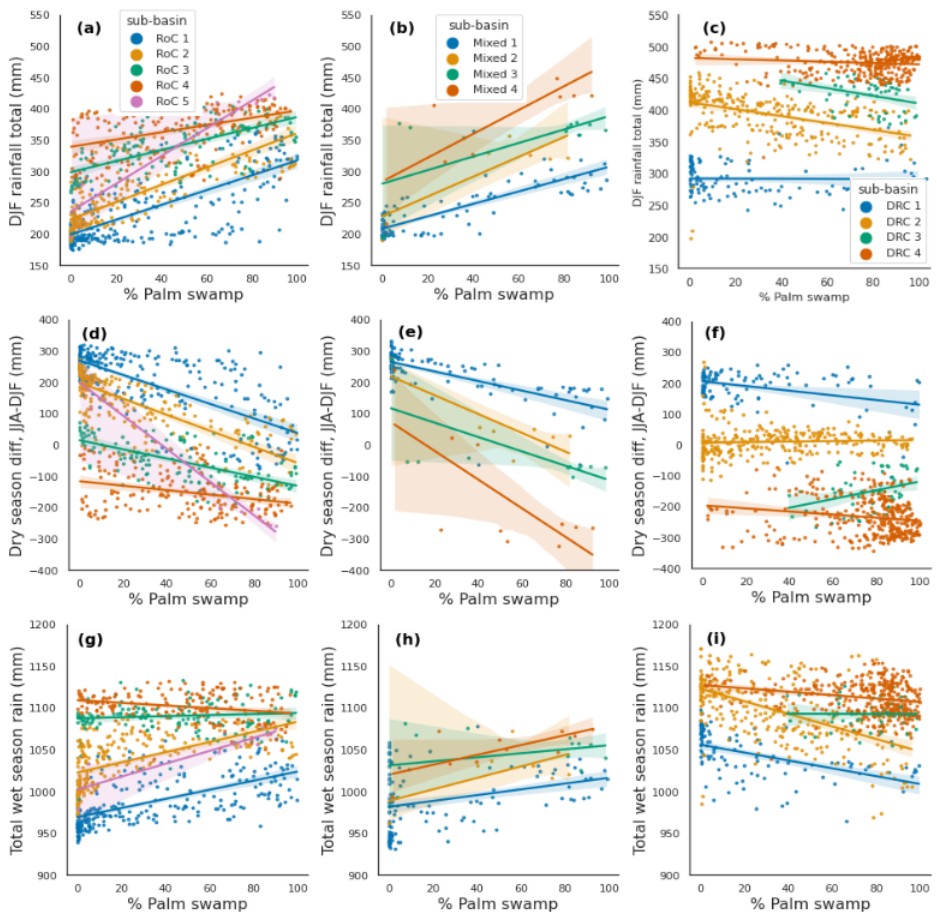

**Figure 7.** Variations in palm swamp composition with rainfall variables used in the regression analysis: first row - DJF rainfall accumulations, second row - dry season rainfall difference (JJA - DJF), and third row - total wet season rainfall accumulation, for the RoC, Mixed and DRC labelled sub-basins. 0% palm swamp is equivalent to 100% hardwood swamp and vice versa.

We also observe greater spatial variation in the dry season rainfall difference than for that between the wet seasons (comparing Figures 8a and b). Additionally, the climatology density plot in Figure 8c shows both higher inter- and intra-basin differences for the dry season difference, with southern sub-basins experiencing a drier December to February, and Northern ones being drier between June and August (Figure 8a) due to the ITCZ position. In contrast, the second wet season (September to November) is consistently wetter than the first across all sub-basins (Figure 8b).





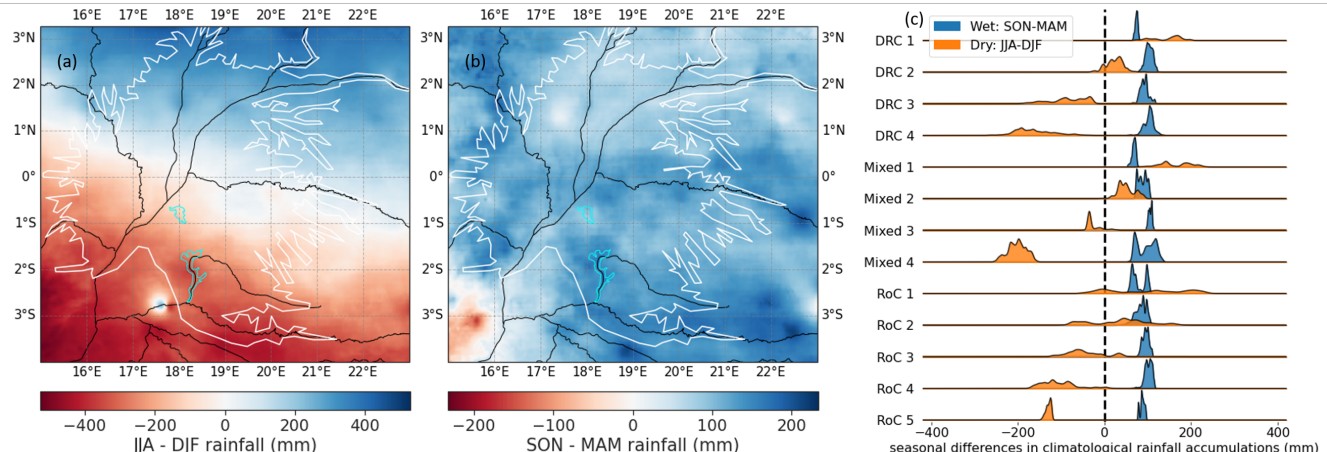

**Figure 8.** The difference in climatological rainfall accumulations between (a) the second (JJA) and first (DJF) dry seasons, and (b) the second (SON) and first (MAM) wet seasons, and (c) a density plot showing histograms of the difference in rainfall accumulation over swamp pixels between the two dry (orange) and wet (blue) seasons.

### 4.1.4 Seasonal temperature correlations with palm swamp composition

The first wet season (MAM) is the warmest (Figure 3d), followed by the first dry season (DJF). For each of the wet seasons, the

increased cloud cover results in increased cloud-radiative forcing at the land surface overall, resulting in higher temperatures, when compared with the preceding dry seasons. We observe significant negative correlations in the linear regressions between the mean seasonal maximum temperature and the palm swamp composition for the first dry season (DJF) and contrastingly, significant positive correlations for the second wet season (SON) for the RoC and Mixed sub-basins (Figure 9). Although there is as much variation in dry season temperatures for the DRC sub-basins, as for those in the RoC and Mixed regions,

there are not significant correlations with palm swamp composition for most DRC sub-basins. The exception is for DRC 2, where we observe a significant positive correlation with the mean DJF maximum temperature, in contrast to the negative correlations observed for the RoC and Mixed sub-basins. Higher temperatures result in increased evapotranspiration and less net water input. This corresponds well with palm swamp presence being negatively correlated with annual rainfall totals for DRC sub-basins.



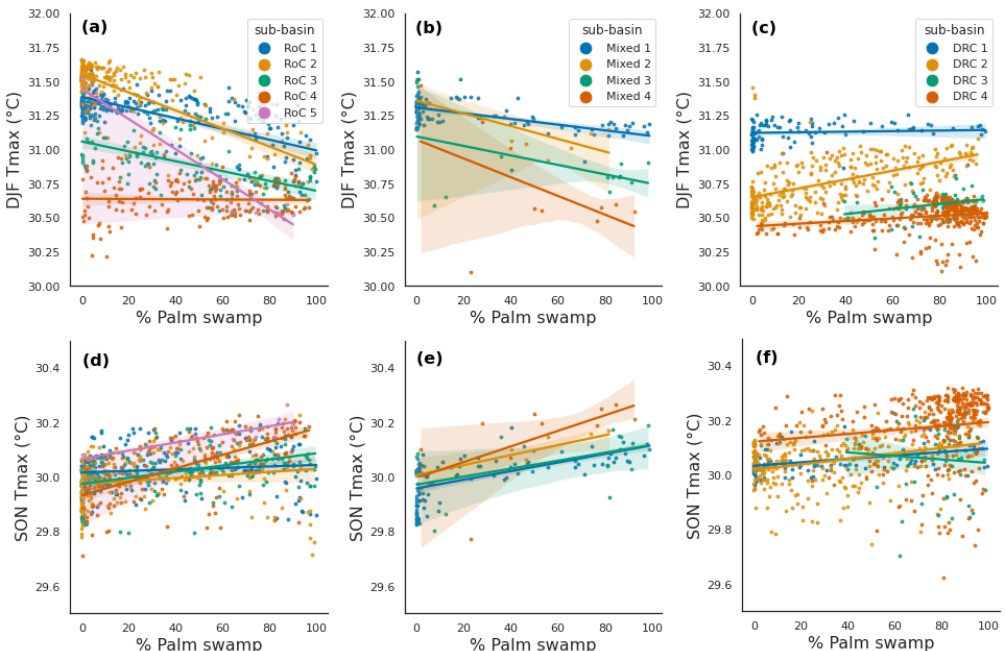

**Figure 9.** Variations in palm swamp composition with the temperature variables included in the regression analysis: mean seasonal maximum temperature for the first dry season for the (a) RoC (b) Mixed and (c) DRC labelled sub-basins, and mean seasonal maximum temperature for the second wet season for (d) RoC (e) Mixed and (f) DRC sub-bsains. 0% palm swamp is equivalent to 100% hardwood swamp and vice versa.

## 4.2   Results from the multi-collinearity analysis used to inform the regression model implementation

Table 2 lists the seven variables and their variation inflation factors (VIF), that we included in the regression model. The elevation and wet season rainfall and temperature variables are sufficiently independent from one another. However, there is structural and data multi-collinearity between the feature variables. For example, structural multi-collinearity exists between our use of the first dry season (DJF) rainfall accumulation and the derived difference in accumulations between the two dry seasons (JJA-DJF), and data multi-collinearity between the DJF rainfall accumulation and mean maximum temperature (Table 2). We decided to include all of them in the final model, despite this multi-collinearity, and refer to the final model significance in relation to these variables as being due to the combined effect of them.

We also determined that seasonal rainfall variables contribute a significant role to swamp vegetation type composition in the RoC, northern DRC, and in some locations which span across the Congo river. We therefore only used the RoC 1 to 5 and the Mixed 1 to 4 labelled sub-basins, which cover these regions, within our final regression analysis. Inclusion of DRC peat swamps located over flood plains would lead to dampening of the relationships we observe between seasonal climatological variables and swamp vegetation type in the inter-fluvial regions. As such, it is preferable to treat the RoC and Mixed sub-basins as largely having a different hydrological regime from those in the DRC.





**Table 2.** Variation Inflation Factors (VIF) and corresponding tolerance calculated between the set of feature variables included in the final beta regression model implementation over pixels with > 70% total swamp composition. VIF > 10 and tolerance < 0.1 indicate multi-collinearity between variables. Features in bold show multi-collinearity.

| Feature variable | VIF - all sub-basins | Tolerance | VIF - modelled sub-basins | Tolerance |
|---|---|---|---|---|
| Elevation | 2.9 | 0.34 | 5.3 | 0.19 |
| Elevation standard dev | 1.4 | 0.73 | 1.1 | 0.93 |
| **DJF total rainfall** | 21.3 | 0.05 | 42.2 | 0.02 |
| **DJF max temperature** | 12.2 | 0.08 | 9.5 | 0.11 |
| **Dry season rainfall difference** | 12.0 | 0.08 | 34.6 | 0.03 |
| Total wet season rainfall | 4.2 | 0.24 | 6.3 | 0.16 |
| SON max temperature | 2.4 | 0.41 | 2.1 | 0.49 |

### 4.3 Comparison of different sets of regression model inputs

The variables included within our three regression model implementations and their statistics are presented in Table 1. The p-values were small for all variables within each model tested. The log-likelihood describes the combined probability of the multiple input variables to predict the dependent variable, and the lower the value the more probable the derived relationship. There is no set optimum range for the log-likelihood statistic as it is specific to each model, however, different parameter combinations can be tested to find those that give the lowest log-likelihood statistic. Although the log-likelihood value was

higher for our final model choice than for the others, this is due to the incorporation of other variables, with additional inherent variability of their own, and we deemed our use of this model to be acceptable due to the significant p-values, higher pseudo-$R^2$ value and low residual values (see Figure 11).

Residual values are calculated by subtracting the predicted value from the observed value. A positive residual indicates that the model under-estimates for that particular data point, and vice versa. The median weighted standardised residual value for

our final model was $-0.06\sigma$ (Table 1), indicating that, overall, the model slightly overestimates the palm swamp composition. In contrast, the rainfall and temperature only models underestimated the palm composition overall. This value was calculated by dividing the residual for each data point by its estimated standard deviation, and then finding the median across all data. It is informative to weight and standardise the residuals as this quantifies them in units of standard deviation, enabling us to more readily identify outliers.

### 335 4.4 Final model summary

Our Beta regression model implementation successfully predicted palm swamp composition across the RoC and Mixed region sub-basins. Figure 10 shows the standardised coefficients corresponding with the beta regression model outputs. These give an indication of the contribution of each regressor to the palm swamp composition. The coefficients for all the independent variables are statistically significant (p-value < 0.05) (see Table 1). The elevation contributes most significantly, followed by

the dry season climatological variables, including the difference between the two dry seasons (JJA-DJF), and the total first dry





season rainfall (DJF). The wet season variables (the second wet season rainfall total and the difference in rainfalll between the two wet seasons) also contribute significantly ($p<0.05$) but to a lesser extent.

Table B1 details the final coefficients, intercepts and sub-basin constants that can be inserted into equation A4, and used together with equation A5 (detailed in Appendix A), to predict the palm swamp composition of any given 0.05° latitudinal x

0.1° longitudinal pixel, given the values for each of the contributing elevation and seasonal rainfall and temperature variables. The corresponding residuals are shown in Figure 11. These are plotted for the 20% of the data points that we reserved within our test dataset. As we had observed with our full dataset, the test data is left skewed due to the overall higher percentage of hardwood swamp composition within our selected set of sub-basins. Overall, the predicted values slightly overestimate the palm swamp composition. At the extremes, there is overestimation where areas are largely saturated with hardwood swamp,

while there is underestimation where palm swamp tends towards saturation. We can see this more clearly by looking at the mapped versus modelled palm swamp composition in Figures 11a and b (over all sub-basins) and 11c (for individual sub-basins). For an ideal model fit, the residuals would have a normal distribution around a zero mean. There is good agreement between the model statistics for the test and training datasets (and also across all data), with similar mean absolute error (MAE) statistics of between 8.6 and 8.9%, root mean square errors (RMSE) of between 14.8 and 15.7% and $R^2$ values of between 0.76

and 0.79 (see Figure 11). As such, we are confident that the model does not over-fit to the combination of feature variables.

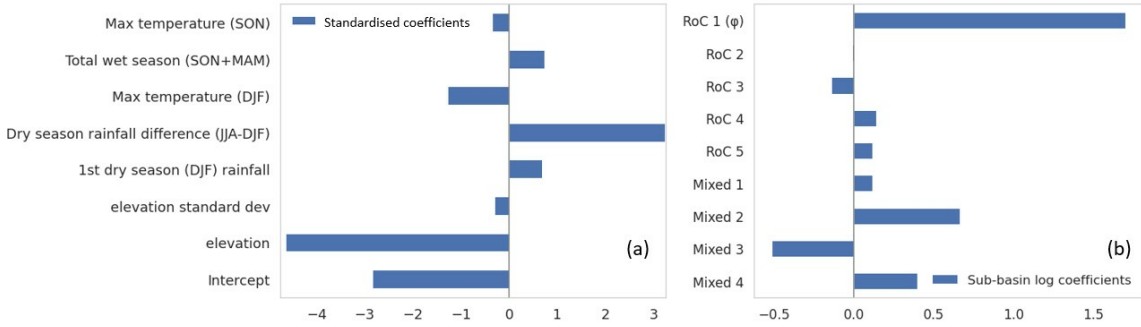

**Figure 10.** Model coefficients for (a) the relative contribution of each standardised independent variable to the calculation of palm swamp composition within each 0.05° x 0.1° pixel (Pseudo-$R^2$ = 0.75) and (b) the categorical sub-basin correction value.

The model most optimally fits to Mixed sub-basin 1 (Figure 11c), for which it has a high $R^2$ value of 0.86, followed by neighbouring Mixed 2 and the more southerly RoC 5 (both with $R^2$ = 0.8 to 0.82). The model also predicts well for RoC 1, RoC 2 and their neighbouring Mixed 3 ($R^2$= 0.76 to 0.78). These are the most northerly sub-basins on the right bank of the Congo river, and experience similar: seasonal rainfall and temperature profiles (Figure 3); elevation ranges; and seasonal

rainfall difference profiles (Figure 8). The model performs least well for sub-basins RoC 3 and Mixed 4 ($R^2$= 0.60 and 0.57 respectively). The Sangha river runs through RoC 3, and has extensive flood plains, while the Mixed 4 sub-basin lies on both sides of the Congo mainstem and likely experiences seasonal river flooding, a component of the net water input that we have not accounted for within our model. Another contributing factor could be that the Congo mainstem is not entirely a blackwater river, as is the case for other tributaries, and therefore has a higher nutrient content, with the leaching of its nutrients into the





floodplains potentially influencing the vegetation composition. Additionally, there are fewer pixels within these sub-basins, and greater associated uncertainty (Figure 11c).

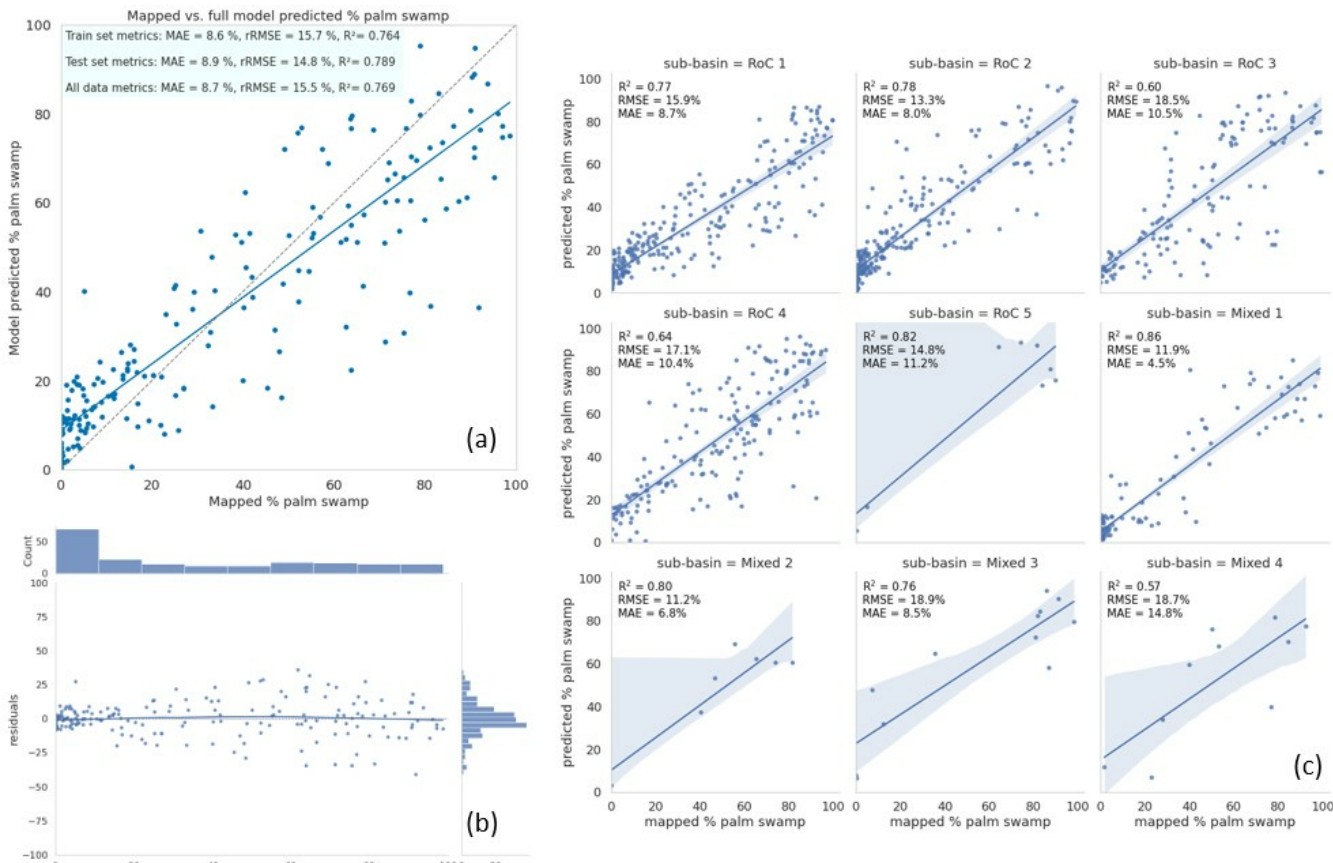

**Figure 11.** (a) Predicted vs. actual palm swamp composition using the test sub-set of points across all sub-basins for which the regression analysis was run. The points represent the data from the 0.05° x 0.1° sub-pixels which contain at least 70% total swamp (palm + hardwood). A linear regression line is included. The overall statistics for Mean Absolute Error (MAE), Root Mean Square Error (RMSE) and $R^2$ are included for each of the training, test and full datasets. (b) Corresponding residuals, with a Locally Weighted Scatter plot Smoothing (LOWESS) line included to describe their trend (the points displayed represent the outputs from applying the regression model to the test dataset), and (c) Individual sub-basin plots of the predicted vs. mapped palm swamp composition using all data points (test and training combined). Summary statistics for each sub-basin are included.

## 4.5  Assessment of outlying predictions

Figure 12a shows the differences between the mapped and predicted values across all the the RoC and Mixed sub-basins. The RMSE for our model predicted palm swamp composition across the full dataset is 15.7%. The negative anomaly outliers are





shown in Figure 12b, and the positive ones in Figure 12c.

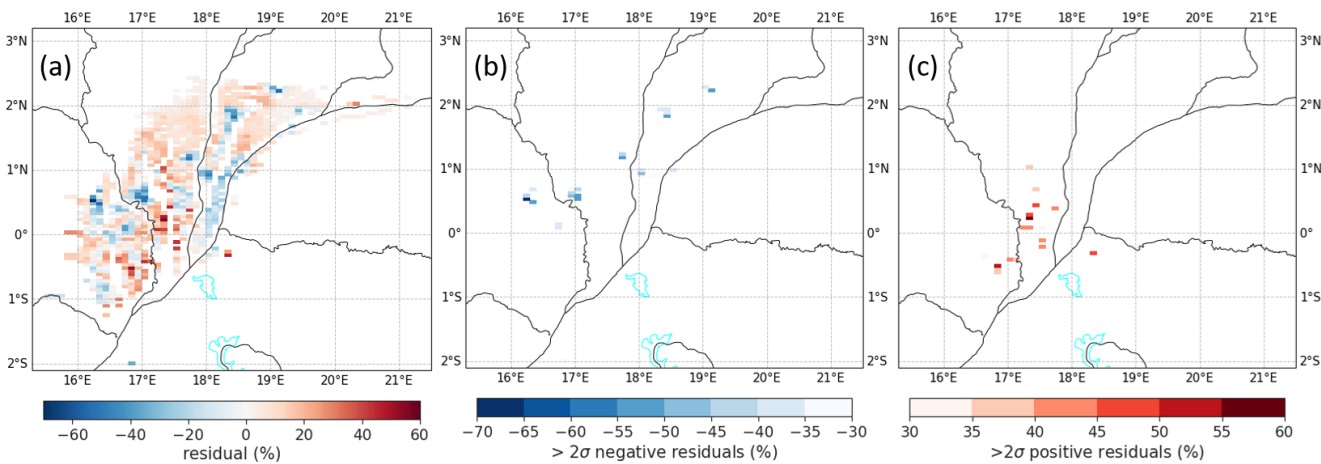

**Figure 12.** The modelled data anomalies from the mapped values for sub-pixels containing greater than 70% total swamp composition for: (a) across all RoC and Mixed sub-basin regions (b) pixels where model underestimation is greater than 2x the standard deviation, $-2\sigma$ (31.4%), from the mapped values, and (c) pixels where model overestimation is greater than $+2\sigma$ from the mapped values.

We observe a concentration of the highest negative anomalies in the 0.5 to 0.8° N latitude, and 16.2 to 17.2° E longitude range. This region passes through RoC sub-basins 2, 3 and 4 (see Figure 2), and spans either side of the Likouala-Aux-Herbes river. There are also negative anomalies observed for the outliers between the Ubangi and Congo rivers, corresponding with

areas located on the right bank of the Ngiri river. Negative anomalies indicate underestimation of the mapped palm swamp composition for those locations, and due to the significant positive correlation between palm swamp dominance and seasonal rainfall accumulations over the sub-basins included in the model, these anomalies may indicate additional water input from the Ngiri river, or from other ground water inputs including sub-surface and surface run-off. The region between the Ubangi and Ngiri rivers is characterised by palm and hardwood swamps, interspersed with permanently flooded wetland (Dargie et al.,

2017; Biddulph et al., 2021). It is therefore likely that this region experiences sufficient net water input to meet the minimum net water input requirement for palm swamps, and if it does receive additional ground water input, then, as with the DRC 1 to 4 sub-basins, this model would not be successful at modelling for these particular locations, as it does not account for all water input sources. Additionally, there are some regions with smaller negative anomalies ($<2\sigma$) at the edges of the Congo river, passing through the Mixed sub-basins 1 to 3 (Figure 12a). The highest positive anomalies are concentrated between 0.5°

S to 0.5° N, and 16.8 to 17.7° E, again corresponding with RoC sub-basins 2, 3 and 4.

We found no significant correlations between the anomalies and the terrain and climatological variables included/considered for inclusion in the regression model. This indicates that our choice of climatology based model features was suitable, as we did not leave out variables that can explain the anomalous predictions. Other possible explanations for the anomalies could be





due to one or some combination of: 1. inter-annual variation in the weather that is not accounted for in our model, e.g. drought
years impacting on the swamp vegetation type composition; 2. additional water inputs from ground water flow or localised river
flooding (e.g. between the Ubangi and Ngiri rivers); 3. variations in evapotranspiration across the region; 4. the longer term
(e.g. centennial/millennial scale) established habitats for vegetation; 5. soil properties, including peat depth, porosity, organic
matter content; 6. the impacts of localised land-use disturbance; 7. errors inherent in the land-type classification map we use
as our ground truth. It is important to note that the vegetation type map data that this study was based on has some inherent
inaccuracy due to it being based on modelled data. Crezee et al. (2022) estimate their peatland distribution model to have a
Matthews correlation coefficient of 80%. Misclassification within the underlying mapped data between palm and hardwood
swamp pixels, or due to the possibility of mixed vegetation types that do not confirm to either of these two main classes, and
are therefore misclassified, may be responsible for the outlying predictions.

### 4.6 Assessing the impact of inter-annual variations in meteorological variables

We were interested in understanding if anomalously dry or wet years contributed to the threshold criteria on rainfall require-
ments for palm swamp dominance in a region. Figure 13 shows the spatial differences in these values for our dry season
modelled variables, in addition to the the annual rainfall totals, for the period 1981 to 2010. We see large inter-annual and
spatial variation for all the rainfall variables, while the standard deviations for the first dry season maximum temperature are
more homogeneous across the basin.

Figures 14a and 14b show the distributions of total annual and DJF rainfall accumulations for the years 1981 to 2010
across all the RoC and Mixed sub-basins. We re-ran our model, with the same inputs as described in Table 1, but using the
drier/wetter year climatologies we calculated (see section 3.7) as alternative inputs to the original CHPclim 1980 to 2009
30-year climatology. We found no significant differences in relationships within the model.

We did however identify a relationship between the ENSO index and weather conditions during the DJF dry season. We
looked at the DJF yearly anomalies from the long-term mean for each sub-basin, and defined anomalously dry years as oc-
curring when the mean yearly rainfall value is less than one standard deviation below the long-term (1981 to 2010) mean
value. There were variations in anomalously dry years between sub-basins, possibly due to the north and southwards migration
extent of the ITCZ. The combined list of years over which at least one of the RoC sub-basins experienced significantly drier
conditions than normal for the DJF season is shown in Table B2 in Appendix B. Here we include the ENSO index from the
DJF period, however if ENSO does impact the severity of dry season rainfall anomalies, then it would likely correspond with
the ENSO index from a few months earlier due to a lag in the translation time from sea surface temperature anomalies to
anomalous atmospheric circulation over West Central Africa.




**Figure 13.** Maps showing the minimum, maximum and standard deviation values for each pixel between 1981 and 2009 for the annual rainfall and three of the dry season variables included within the regression model - first dry season rainfall accumulation, difference in rainfall between the two dry seasons, and the mean of the monthly maximum temperatures across the first dry season (DJF).




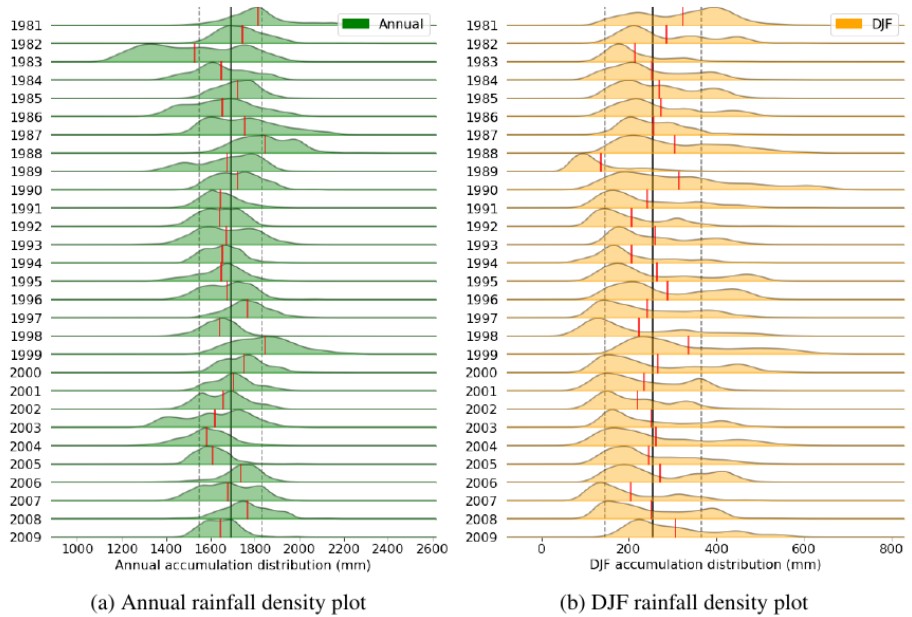

(a) Annual rainfall density plot        (b) DJF rainfall density plot

**Figure 14.** Annual and first dry season yearly (1981 to 2009) rainfall distribution density plots, across all RoC and Mixed sub-basins included in the regression analysis. Over-plotted are the mean values over all years (black line), the standard deviation over all years (black dashed line) and the mean values for the yearly distributions (red lines).

# 5 Discussion

Our results show that elevation and climatological rainfall and temperature variables contribute significantly to determining
where palm or hardwood swamp vegetation types dominate within the Cuvette Centrale peatland complex. Here, we discuss the possible reasons for these relationships and their implications.

## 5.1 The significance of annual and wet season rainfall on swamp vegetation type

The significant positive correlations we observed between annual rainfall and palm swamp composition for all RoC, and the Mixed 1 and 2 sub-basins indicate similar hydrological mechanisms could be playing a role within the inter-fluvial (largely
rain-fed) peat swamps of these sub-basins. In contrast, for Mixed 3 and 4, and DRC 3 sub-basins, we observed no significant correlations with total annual rainfall, and low spread in rainfall totals. This indicates that the total annual net-water input across all sub-pixels within these sub-basins lies within a range conducive to palm swamp dominance, and that other variables, including topographical and seasonal climatological, may be responsible for delineating where palm or hardwood swamp can exist as the dominant peatland swamp type. It is important to note that all Mixed sub-basins are situated along the Congo river
mainstem and are also susceptible to receiving additional water from riverbank overflow. Regional palm swamp composition has a significant and, contrastingly, negative correlation with annual rainfall totals for DRC sub-basins 1, 2 and 4, likely due to





a combination of their location in flood-plains, and that they receive higher rainfall input (up to ~200 mm extra per year) than elsewhere in the Cuvette Centrale.

We also observed significant saturation of hardwood swamp (corresponding with 0% palm swamp in Figures 6a-c) across all annual rainfall totals for: all RoC, Mixed 1, and DRC 1 and 2 sub-basins. This may be a result of long-term historical establishment of hardwood tree varieties in these regions, but could possibly also be due to the localised inter-play of additional hydrological inputs from the river system or surface/sub-surface run-off, not accounted for within this study. As such, peatland within these areas may receive net water input outwith an acceptable range for palm swamp growth.

      Within our model, the total wet season rainfall accumulation was found to contribute significantly to swamp vegetation
type composition, albeit with a smaller impact than for the combined dry season rainfall variables. For RoC and Mixed sub-basins there is a positive correlation between the palm swamp composition and the total wet season rainfall, with the greatest contribution from the second wet season (SON). In contrast, total wet season rainfall contributes more significantly to palm swamp composition than dry season rainfall for DRC sub-basins 1, 2 and 4 (Figure 7i). As such, the addition of other feature variables, e.g. water-table depth, would be required before the same regression model could be used for sub-basins to both the
left and right hand sides of the Congo river.

## 5.2   Optimal net water input requirements for palm swamp dominance

Both palm and hardwood swamp vegetation can be found across the extent of the Cuvette Centrale (Figure 4), but the areas in which they each dominate are observably distinct, with hardwood swamps dominant in the north and around the peripheries of the peatland complex, while palm swamp dominates in the central, inter-fluvial regions to the right of the Congo river, and also
in the south, to the left-bank of the Congo river. Palm swamps dominate in regions of the RoC and northern DRC (between the Ubangi and Congo rivers) which receive higher rainfall totals than for the neighbouring hardwood-dominated swamp regions (Figure 3). However, the RoC and northern DRC sub-basins receive less rainfall than the DRC sub-basins that lie to the left of the Congo river, and in contrast to the pattern for RoC and Mixed sub-basins, DRC regions which receive the highest annual rainfall totals are dominated by hardwood swamp. Interestingly, the linear regressions for sub-basins located on the left (DRC)
and right (largely RoC/Mixed) banks of the Congo river converge towards a similar range of annual rainfall totals as palm swamp composition tends towards 100% (comparing Figures 6a, b, and c, with positive correlations for most RoC/Mixed sub-basins, and negative correlations for most DRC sub-basins). And, given that regions receiving the highest rainfall totals in the DRC seem to favour hardwood swamp tree varieties, there is support for the existence of an upper limit on the net water requirement for palm swamp dominance.

If we assume that the RoC inter-fluvial sub-basins are largely rain-fed, then we can approximate a minimum annual rainfall requirement for palm swamp dominance from the linear regression for RoC sub-basins 1 to 5 as lying somewhere between ~1640 and 1740 mm (Figure 6a). Annual rainfall totals below this range tend towards corresponding with hardwood swamp trees dominating. From the linear regressions for DRC sub-basins 1 to 4, we derive that such an upper limit could lie in the range of 1800 to 1900 mm (taken from the range of points where palm swamp presence reaches 50%). However, there is less





confidence in this upper limit as there is some uncertainty about the role of additional hydrological mechanisms, including river flooding, that may contribute to the net water input for the DRC sub-basins.

Possible reasons for such positive correlations corresponding with rain-fed locations in the RoC and northern DRC (north of the Congo river), and for the possibility of there being an optimal range of net water input for palm swamp dominance are:

i. There could be a minimum net water input requirement for palm swamp that is greater than that for hardwood swamp.

ii. Palm swamp regions which do not receive sufficient additional water input from run-off or river flooding to contribute to the net water requirements for their growth, may require an increased amount of water input directly from rainfall. This could explain the difference in the sign of the relationship with annual rainfall between the RoC sub-basins (positive, see Figure 6a), and those that lie south of the Congo river in the DRC (negative, see Figure 6c), where there are higher rainfall totals.

iii. Dargie et al. (2017) observed that the roots and buttresses of hardwood trees in the Cuvette Centrale are commonly adapted to wet conditions, including being stilted and having aerial roots, as well as buttresses which enable them to stabilise with shallow roots. This may explain why hardwood trees adapt well to regions of the Congo Basin that experience higher levels of rainfall, outwith the optimal water requirements range for palm swamp dominance.

iv. Such a range of net water input would likely be a proxy for the impacts of water-table level on the ability of *Raphia*
*laurentii* palms to become established under different inundation conditions. For example, a minimum or maximum root wetness duration. As such, dry season rainfall totals and duration could be of particular importance in relation to the provision of a minimum root wetness threshold within the inter-fluvial RoC palm swamps, which receive less net water input than those in the DRC.

v. Conversely, in the floodplain DRC swamps, which receive sufficient water input to always meet the minimum threshold
for palm swamp dominance, there may be some level of maximum inundation, or length of continuous inundation, which *Raphia laurentii* cannot support.

### 5.3 The impact of dry season rainfall totals on swamp vegetation type

Additionally, the first dry season rainfall totals (DJF) are positively correlated with palm swamp composition for RoC and Mixed sub-basins (Figures 7a and b), while the second dry season rainfall totals are negatively correlated. As such, the differ-
ence in rainfall totals between the two dry seasons is significant (Figures 7d and e), while the total dry season rainfall is not. This provides additional support to there being an optimal range of water input, within which *Raphia laurentii* palms have a competitive advantage over hardwood trees. The reasons as to why this is requires some further exploration. We hypothesise that this advantage may be possible if *Raphia laurentii* palms have faster growth rates than hardwood trees, such that when the rainfall conditions are suitable for both swamp vegetation types to be present, the palm swamp can more quickly get estab-
lished to the point of leaf development and photosynthetic activity. It could also be due to specific adaptations common to the





*Raphia* genus of palms, e.g. the presence of pneumatophores allowing increased oxygenation at the roots under flooded conditions, as with *Raphia hookeri* (Jeník et al., 1967) and *Raphia taedigera* (Wright et al., 2013; Girkin et al., 2018). Literature on the physiological characteristics of *Raphia laurentii* is currently lacking for further conclusions to be drawn on this. Field observations of the physiological and morphological characteristics specific to *Raphia laurentii* would help to establish the

underlying reasons for the differences in optimal climatological conditions between the palm and hardwood swamp vegetation types present in the Cuvette Centrale.

In our final model implementation, inclusion of elevation as a feature variable solved for some of the lack of predictive ability with our rainfall-only model, but at the expense of reduced significance of the dry season rainfall variables. There was also a reversal in the order of importance between the dry season rainfall features due to multi-collinearity. We concluded that some

combination of the dry season rainfall and temperature variables contributes significantly to regional palm swamp composition. The regression model results support our hypothesis that areas where palm dominates benefit from higher dry season rainfall totals. Additionally, there is collinearity between the dry season variables and the elevation. Although our final model more accurately predicts palm swamp composition, some additional contribution of the dry season rainfall may be masked within the modelled significance of the elevation variable.

**5.4   Why is elevation of significance when modelling swamp type composition?**

As elevation increases, palm swamp composition decreases, giving way to hardwood swamp dominance. There is some uncertainty as to why the elevation is of such significance. We surmise this could be due to some combination of: 1. surface and sub-surface run-off, providing additional water input at lower elevations; 2. decreases in temperature with altitude; 3. the impact of variations in cloud cover with elevation on temperature and humidity; 4. geomorphological reasons.

The elevation ranges within each sub-basin are relatively low, with the highest range (for the subset of sub-basins we included in the model) of 155 m for RoC 4, and an average range over the nine sub-basins of ~105 m. If we assume the moist tropical lapse rate to be ~ -0.6°C 100 m$^{-1}$ (Grab, 2013; Johnson et al., 2016), then this translates to an average range in temperatures due to altitude of 0.63°C (min for RoC 5: 0.27°C, max for RoC 4: 0.93°C). However, the seasonal temperature pattern does not closely follow the topography (comparing Figures 2 and 3d), except in the peripheral regions of the Cuvette Centrale where the

topography is significantly higher than that of the modelled swamp locations. At the lower altitudes where peat swamp exists, temperature changes with altitude are likely buffered by the blanketing effect of cloud cover.

Across the RoC and Mixed sub-basins, and below 320 m elevation, neither the elevation, nor the variance in elevation over the 0.05° x 0.1° sub-pixels, contribute to differences in swamp composition, although, over all elevations, these variables contribute significantly. This may be attributable to run-off from higher elevations having contributed to the pooling of water

comparably for swamp regions below 320 m, such that the pattern of total net water input is more clearly reflected in the pattern of rainfall totals in rain-fed regions.

Although the elevation standard deviation was included in our final regression model, it has the highest p-value of all feature variables ($p = 0.035$), and was of only border-line significance ($p \sim 0.05$) for some of the other random-seed test implementations of the model. It also contributes minimally to determining the swamp type composition when compared with





the other variable contributions (coefficient = -0.3, Table 1). We had originally included this variable to account for regions of steeper terrain where run-off may occur, however its low modelled significance may indicate that run-off between pixels cannot be well accounted for at the spatial scale our model has been implemented.

There are levees between the Congo basin rivers and swamps (Campbell, 2005), which control to some extent river flooding over peatland areas. However, areas that are both situated close to the river system and which have low elevation standard deviation, may be prone to experiencing more frequent flooding (and sediment deposition) due to their lack of a high enough levee when river levels tend towards their maximum. There is low elevation standard deviation observed across the Mixed sub-basins, which span the Congo river, and this may indicate their susceptibility to flood water inputs. In contrast, the correlation between elevation standard deviation and palm swamp composition was higher for the southern DRC sub-basins 2, 3 and 4, indicating an increased likelihood of surface/sub-surface run-off inputs contributing to total net water input for low-lying swamp locations, in addition to flooding from tributaries. The significant contribution of both elevation and elevation standard deviation in these DRC sub-basins may also imply that swamp regions receive additional water input from run-off, and potentially also as a result of flooding from smaller tributaries with lower levees, rather than from flooding from the lower elevation major river tributaries.

The differences in significance of the relationship between palm swamp composition and elevation standard deviation, between sub-basins located on the left and right sides of the Congo river, points towards diversity in hydrology across the region. Further investigation of the spatial and temporal evolution of hydrology is required across the peatland complex.

## 5.5 Impact of temperature on swamp vegetation type

We also observed that palm swamp dominates in regions with higher mean annual maximum temperatures (Figure 3c). Temperature and precipitation are highly correlated due to the influence of cloud cover on downwelling radiation incident on the land surface. There are significant correlations between seasonal maximum temperature climatology and palm swamp composition, including for the DRC sub-basins which we did not run our model over (Figure 9). Higher temperatures lead to increased evapotranspiration, and less soil water retention. If all regions within these sub-basins, situated on the left side of the Congo river, receive sufficient rainfall inputs, and potentially additional water input from river flooding, surface or sub-surface flow, then distinguishing their suitability for palm or hardwood swamp dominance would not be possible by consideration of seasonal rainfall inputs alone. And this does seem to be evident in the low correlations between seasonal rainfall and palm swamp composition for these DRC sub-basins. However, temperature and evapotranspiration are positively correlated, and the use of seasonal net water input as a variable in our model would allow for corresponding variations in rainfall and temperature to be better accounted for. Additionally, the use of a spatio-temporal flood depth map as an additional variable within the model would enable better attribution of the combined impacts of temperature and rainfall on swamp vegetation type dominance across all regions of the Cuvette Centrale. Improved understanding of these combined impacts would enable improved understanding of the regional climate change impacts on swamp vegetation type distribution.



## 5.6 Rainfall-runoff mechanisms

A bowl-like topography is present in the Cuvette Centrale (Figure 2), with an oval of steep upland topography giving way to an extensive lowland region of low-varying slope. By comparison with Figure 1, the correspondence between elevation and palm

swamp dominance can be observed. This is especially the case in the 295 to 315 m elevation range. Within the set of variables we investigated, we calculated elevation as having the highest Pearson correlation ($\rho$) with palm swamp composition ($\rho$=-0.80), while the height above nearest drainage basin (HAND), although significant, did not have nearly as high a correlation ($\rho$=-0.34). The floodplain peatlands of the DRC sub-basins which are situated to the left-bank of the Congo river receive additional water inputs from river flooding and potentially ground water flow. The topography that borders the DRC peatlands is steeper than

for the RoC peatlands. Inclines to the south and south-west of the DRC sub-basins give way to the floodplains, and the bowl-like topography of the Cuvette Centrale may direct additional hydrological input to the peatlands via less obvious surface, and potentially sub-surface, water channels, in addition to the main tributaries. Run-off contributions may add significantly to the peatland's net water input, in the DRC particularly. A rainfall-runoff model, ideally with field measurement inputs, would be required to assess the significance of run-off contributions to the lowland DRC peatland regions located in the sub-basins

bounded by the Congo river. Previous rainfall-runoff modelling studies (Tshimanga and Hughes, 2012; Tshimanga et al., 2011) have identified surface and sub-surface responses to rainfall, including contributions to the storage of water within the wetlands. Kabuya et al. (2020) cite a lack of in-situ hydrological information to predict regional rainfall-runoff interactions across the Congo river basin.

There may also be non-linear dynamics at play, with regions of increased soil moisture (e.g. in the water-logged peatland

regions) having a positive feedback on rainfall patterns, resulting in enhanced evaporation and cloud formation. This could be a potential explanation for the large region of enhanced annual rainfall we observe in Figure 3a. Such a feedback could be explored further by assessing the relationship between the temporal evolution of land water storage and rainfall accumulation.

## 5.7 Are the derived relationships with seasonal climatological variables unique to the Cuvette Centrale?

Palm swamp is able to dominate in regions of Peru with much higher rainfall totals than experienced anywhere in the Cuvette

Centrale (Marengo, 1998), while there appears to be an upper threshold on net water input requirements for the *Raphia laurentii* palm swamp type to be able to dominate in the Cuvette Centrale. However, this is most likely due to each peatland complex supporting different species of palm, with different physiological adaptations to the climate. The relationships we have derived for *Raphia laurentii* may be unique to the Cuvette Centrale. This would require further investigation in a future study, taking into account additional swamp vegetation physiological processes and features.

## 5.8 Climate change impacts on peat swamp vegetation type

There is currently high uncertainty in future climate change impacts on rainfall over the western Central Africa region (Niang et al., 2014). Tshimanga and Hughes (2012) determined that the impact of climate change on total run-off from the Congo Basin is likely to be minimal under the A2 climate change scenario, which is one of the higher end climate change scenarios





presented by the IPCC within their Special Report on Emissions Scenarios (SRES) (Di et al., 2011). However, their study also identified a decrease in future run-off for the northern Congo basin. The Coupled Model Intercomparison Project Phase 6 (CMIP6) projects that extreme rainfall amounts will increase over the Central Congo Basin by between 10 and 35% on the wettest day of the year, corresponding, respectively, with when global warming levels of between 1.5 and 4°C above the long-term average will be reached (Ariais et al., 2021). Within the latest IPCC report, Pörtner et al. (2022) projected increases in average annual rainfall and heavy rainfall events (calculated as maximum accumulations over five-day periods), while there

are projected decreases in drought frequency, intensity and duration for global warming of 1.5 to 3°C above the pre-industrial average temperature. The models do not confidently agree that these projections show changes beyond natural variability.

Within our model, the dry season rainfall and temperature contributions were more significant than those from the wet season due to the much higher spatial variability in dry season rainfall. There is large spatial (inter-basin) variance (Figure 3a) and large inter-annual variance (Figure 13c) in rainfall accumulations. Our results indicated that anomalously dry periods correlate

with El Niño events and it would be interesting to investigate this relationship further and determine if swamp vegetation type is historically influenced by El Niño, or by other events which impact atmospheric circulation (e.g., the Indian Ocean Dipole). Our limited inter-annual analysis was not sufficient to assess this, and further understanding is required of the larger scale atmospheric processes which drive inter-annual variability in rainfall over the Congo Basin. A longer term study, e.g. from the early 1900s, could be done using reanalysis data to better assess inter-annual variations in rainfall, and their relation with swamp

vegetation type dominance, and to assess if anomalously low rainfall seasons provide the delineating conditions that determine the areas of the Cuvette Centrale where palm swamp is present. Our investigation suggests that there is an optimal range of rainfall totals under which palm swamp dominates, and outwith which hardwood swamp tree varieties dominate. Given the relationships derived within our regression model implementation, future changes in rainfall duration, intensity or seasonality, as well as the projected temperature increases, could impact on the likelihood of palm or hardwood swamp vegetation types

dominating in a particular region of the Cuvette Centrale. Improved certainty in projections of seasonal rainfall change under the different climate change scenarios would enable us to better assess how the distribution of swamp vegetation type could be affected.

## 5.9 Other considerations and opportunities for future research

Another contributing factor to the propagation of swamp vegetation could be peat depth, although the mechanisms of cause and

effect remain uncertain, and it is not clear if pre-existing peat depth would impact swamp type, or if swamp type would lead to differences in peat accumulation, or if there is a coupled dynamic between the two. An initial investigation by Dargie et al. (2017) measured peat depth for palm and hardwood swamps, with no significant difference found between the two. Peat depth has recently been more comprehensively mapped for the Cuvette Centrale region (Crezee et al., 2022), with slightly higher peat depth and carbon density predicted for palm swamp. This is likely because palm swamps are often located in the deeper

interiors of peatland areas. A future study could use this dataset to investigate if correlation between swamp vegetation type and peat depth exists over the extent of the Cuvette Centrale. Vegetation type and its underlying soil properties also play a role in the spatial and temporal evolution of inundation. Peat organic matter properties have been found to vary by vegetation type (Girkin





et al., 2020; Upton et al., 2018; Sjögersten et al., 2011) due to variations in plant litter-fall and chemistry. Resulting variability in peat composition and porosity can affect the rate at which the soil becomes water-logged. Differences in water-logging

impact net water availability to swamp vegetation. It would be interesting to investigate further the relationships between peat composition, porosity, and vegetation type, taking into consideration net water input. Additionally, it would be interesting to consider lagged correlations between net water accumulations and swamp vegetation type composition, and also differences in the length of the wet and dry seasons to the north and south of the equator. Using fixed three month seasonal averaging periods has limitations in that it is not fully representative of variations in rainfall seasonality across the whole basin.

**6  Conclusions**

We have used a beta regression model to assess the contribution of terrain and seasonal climatological variables to palm swamp type composition in the Cuvette Centrale peatland complex. We found significant relationships with elevation and dry season variables (rainfall and temperature) in particular, with additional significant contributions from the total wet season rainfall and wet season maximum temperature variables. The likelihood of palm-dominated vegetation increased with: decreasing

elevation; increasing difference in rainfall accumulation between the two dry seasons; decreasing first dry season temperature; increasing total wet season rainfall accumulation; increasing first dry season rainfall accumulation; and decreasing second wet season temperature, in that order of importance (all p<0.05). Due to multi-collinearity between the dry season climatological variables, we can more simply say that palm swamp dominance primarily varies as a function of elevation and a combination of primarily dry season, and to a lesser extent, wet season climatological variable contributions. Our model successfully predicts

the percentage palm swamp composition (overall $R^2$ ~ 0.77) for sub-basins in the RoC, DRC (north of the Congo river), and for those which span across the Congo river. The higher levels of rainfall input, and the possibility of additional ground water inputs over the rest of the DRC, mean that our model cannot be used to predict swamp vegetation type composition over this region without the inclusion of additional variables accounting for the total net water input at each pixel. Further field-based investigation is required to confirm the extent of hydrological input from the river system and ground water flow across the

Cuvette Centrale, but the results we discuss within this study support the contribution of additional water inputs from run-off, particularly in the DRC, with cross-regional differences in the sub-basin swamp type composition response to rainfall inputs.

Our results indicate that palm swamp dominates within an optimal range of net water input totals, outwith which hardwood swamp trees can dominate over a much wider range of annual rainfall accumulations. This implies that palm swamps have an evolutionary advantage/adaptation that enables them to dominate over hardwood swamps within a certain range of net water

input accumulations.

*Code and data availability.*  The land type map data that underlies this study is an output from the CongoPeat project, and will be made open-source following project completion. Additionally, jupyter-notebooks with the Python code for all stages of this investigation are available at: https://github.com/SelenaGeorgiou.



## Appendix A: Regression model description and setup

We applied Z-score normalisation to each variable using:

$$Z = \frac{x - \mu}{\sigma} \tag{A1}$$

where x is the original value, $\mu$ is the population mean and $\sigma$ the standard deviation.

     The beta regression model uses a logistic-link function which converts the dependent variable, palm swamp composition, to

a logistic input. We divided our training data variable, percentage palm swamp composition, by 100 to express it as a fraction. We also had to convert fractions that were exactly 0 and 1, to 0.000001 and 0.999999 respectively, as otherwise these values would tend to $\pm\infty$ when converted to logistic input using the following equation:

$$PS_L = ln\left(\frac{PS}{1\text{-}PS}\right) \tag{A2}$$

     The linear regression was then applied to the data using the logistically converted dependent variable ($PS_L$) and the selected

independent variables: elevation, standard deviation of the elevation, dry season rainfall difference, first dry season rainfall, total wet season rainfall, mean maximum temperature for the first dry season, and mean maximum temperature for the second wet season. Additionally, the sub-basin value (RoC 1 to 5 and Mixed 1 to 4) was entered into the model as a categorical variable. Two equations are described when using beta regression. The first takes into account the intercept and feature variable coefficients output from the regression, and is expressed as:

$$PS_L = \beta_0 + \beta_1 x_1 + \beta_2 x_2 + ... \tag{A3}$$

where $\beta_0$ is the intercept and $\beta_1$, $\beta_2$ etc. are the derived coefficients for the independent variables.

     In our specific case, this evaluates to the linear-predictor equation:

$$PS_L = \beta_0 + \beta_1 * elevation + \beta_2 * elevation\_stdev + \beta_3 * (JJA - DJF) + \beta_4 * DJF$$
$$+ \beta_5 * (MAM + SON) + \beta_6 * DJF\_Tmax + \beta_7 * SON\_Tmax \tag{A4}$$

     The mean expected proportion ($\mu$) of palm swamp is then calculated by applying the inverse link function to the linear-predictor (eq.A4) output :

$$\mu = \frac{1}{1 + e^{-PS_L}} \tag{A5}$$

where $\mu$ is the fractional palm swamp composition within a given 0.05° x 0.1° pixel. The regression with non-z-score coefficients returns log-odds of each independent variable's contribution to the prediction.





Our implementation of beta regression used a logistic link function for the independent features, and a log-link for the categorical feature (the sub-basin). The log-link equation is used to account for variation in the predicted proportions resulting from the sub-basin categorical variable (Cribari-Neto and Zeileis, 2010):

$$\phi = e^{\gamma_0 + \gamma_1 z_1 + \gamma_2 z_2 + \gamma_3 z_3 + \dots} \tag{A6}$$

where $\gamma_n$ are the $\phi$ precision coefficients, and $z_n$ are the sub-basin regressors (which each have a binary value of 1 for the chosen sub-basin and 0 otherwise). The higher the values of $\phi$, the lower the variance for a given mean expected proportional value of the palm swamp composition, $\mu$.

This second equation for the $\phi$ precision does not affect the first equation outputs (eq.A5), however it does allow for the variance due to sub-basin differences to be accounted for:

$$Var_{sub-basin} = \frac{\mu(1-\mu)}{1+\phi} \tag{A7}$$



## Appendix B: Additional tables

**Table B1.** Model beta and sub-basin $\phi$ precision coefficients corresponding with equations A4 and A5. The feature variable coefficients were calculated using a beta regression mean model with logit-link, and the Phi coefficients for the categorical sub-basin variable were calculated using a precision model with log-link. Either the standardised or non-standardised coefficients can be used to predict the palm swamp composition given the listed feature variable values for each sub-basin. The statistics for each sub-basin are also detailed, including the Mean Absolute Error (MAE), the Root Mean Square Error (RMSE) and the $R^2$ value.

| Feature variable coefficients | Coefficient | Z-score value | Non-standardised value |
|---|---|---|---|
| Intercept | $\beta_0$ | -2.832110 | 205.3069044 |
| Elevation | $\beta_1$ | -4.639456 | -0.2475632 |
| Elevation standard deviation | $\beta_2$ | -0.298973 | -0.0470500 |
| Dry season rainfall difference | $\beta_3$ | 3.230571 | 0.0146008 |
| first dry season rainfall (DJF) | $\beta_4$ | 0.672975 | 0.0066924 |
| Total wet season | $\beta_5$ | 0.726320 | 0.0133626 |
| DJF mean Tmax | $\beta_6$ | -1.284508 | -3.1527621 |
| SON mean Tmax | $\beta_7$ | -0.353391 | -1.5315068 |

| Sub-basin category constants | | value | MAE | RMSE (%) | $R^2$ |
|---|---|---|---|---|---|
| RoC 1 ($\phi$ precision) | $\phi$ | 1.7010966 | 8.7 | 15.9 | 0.77 |
| RoC 2 | c | -0.0127231 | 8.0 | 13.3 | 0.78 |
| RoC 3 | c | -0.1422987 | 10.5 | 18.5 | 0.60 |
| RoC 4 | c | 0.1388437 | 10.4 | 17.1 | 0.64 |
| RoC 5 | c | 0.1153412 | 11.2 | 14.8 | 0.82 |
| Mixed 1 | c | 0.1178069 | 4.5 | 11.9 | 0.86 |
| Mixed 2 | c | 0.6638891 | 6.8 | 11.2 | 0.80 |
| Mixed 3 | c | -0.5135390 | 8.5 | 18.9 | 0.76 |
| Mixed 4 | c | 0.3988342 | 14.8 | 18.7 | 0.57 |

**Table B2.** Years in the 1981 to 2010 period where some/all of the RoC and Mixed sub-basins experienced anomalously dry conditions during the first dry season (DJF), and their association with El Niño/La Niña events.

| Year | El Niño (EN) or La Niña (LN)? | ENSO index during DJF | Duration of EN/LN |
|---|---|---|---|
| 1983 | EN | 2.2 | Apr 1982 to Jun 1983 |
| 1987 | EN | 1.2 | Sept 1986 to Feb 1988 |
| 1989 | LN | -1.7 | May 1988 to May 1989 |
| 1992 | EN | 1.7 | May 1991 to Jun 1992 |
| 1994 | - | 0.1 | - |
| 1998 | EN | 2.2 | May 1997 to May 1998 |
| 2007 | EN | 0.7 | Sept 2006 to Jan 2007 |



*Author contributions.* S.G. conceived the study, developed the methodology and code, performed the analysis and wrote the manuscript. E.T.A.M. supervised the research and contributed to the methodology. B.C. provided the land type map data that formed the basis of this study. E.T.A.M., P.I.P., B.C., G.C.D., S.S., N.T.G., and S.L.L. reviewed the manuscript and provided suggestions that were incorporated into the final version. B.C., G.C.D., C.E.N.E., O.E.B., J.K.T., P.B., J.B.N.N., N.T.G., Y.E.B., S.A.I., and S.L.L. were involved in the collection of field site data used to develop the land type classification map that underlies this study.

*Competing interests.* The authors have no competing interests to declare.

*Acknowledgements.* Funding for this research was provided by NERC through an E4 DTP studentship (NE/S007407/1), and the CongoPeat grant (NE/R016860/1; https://congopeat.net) which provides the co-authors with full or partial financial support. We would like to thank Achim Zeileis who helped us to understand the phi-precision calculation involved within the Betareg model (described in Cribari-Neto and Zeileis (2010)) through their comprehensive answer to a question asked in a statistics forum. Thanks to all the software developers of open-
source Python and R packages that enabled us to carry out this research efficiently, and thanks also to the developers of openly available meteorological and terrain data sets used within this study, and to the organisations that have provided their satellite data under open source terms for use within the development of these products.





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
