# Peer review of "What determines peat swamp vegetation type in the Central Congo Basin?"

_EGUsphere, 2022_

## Author Comment (AC1)

**Response to reviewer 1**

We would like to thank reviewer 1 for taking the time to review our manuscript and for their useful comments, which will help to shape this into an improved paper. We provide here our responses to each comment made and how we will modify the manuscript as a result. The reviewer's comments are in bold, and our response in normal text.

**Row 30: maybe reference to Fig. 1 could be on the row where CC is mentioned for the first time (27)**
Yes, we agree - we changed this as suggested. Thank you.

**Row 34: increases the carbon stock; does this refer to situation that Cuvette Centrale wouldn't exist? Wouldn't this be easier to say as a proportion of the total carbon stock?**
We agree that this would be a better way to express it, and have changed it in the text accordingly. Thank you for the suggestion.

Prior to the Dargie et al. (2017) study, it was understood that there was peat in the Cuvette Centrale, but the extent of it hadn't previously been mapped, and its contribution to the world's tropical peatland carbon stock had been greatly underestimated. Crezee et al. (2022) estimate that it could contribute a total of 29.0 petagrams to the world's belowground tropical peatland carbon stock, equivalent to 28% of the best estimate of tropical peatland total belowground carbon stock. We will replace the reference in row 34 with this proportional reference from Crezee et al. (2022).

**Row 91: as Crezee et al. (2022) land classification map is a data of high importance in this paper, it would be fair to describe a bit of how it was constructed (as well as acknowledging its potential sources of error, which may also affect on e.g. detected anomalies)**
Agreed, thanks. We will add a further brief description of the random forest regression model they used to classify the land types, and how the accuracy of this map was assessed against ground-based forest inventory data. This will be added to the current row 40, where we first describe how we will use Crezee et al. (2022) data.

We use the Crezee et al. (2022) land type map as the current best representation of the Cuvette Centrale's land type distribution, but it does have inherent errors. They used Balanced Accuracy (BA) as a metric for assessing the accuracy of their derived peatland extent, and calculated a BA of 91.9% (95% CI, 90.2–93.6%). Crezee et al. (2022) estimate their peatland distribution model to have a Matthews correlation coefficient of 80%.
In the current manuscript we mention on line 393 that inherent errors in the land type map may have contributed to the anomalies we observe in our model. We will add quantification of these errors to the manuscript, and make it clearer that these errors will propagate into our model's estimates.

**Rows 167-171: I'm not totally convinced of the use of STD in this context; it kind of reflects the uncertainty or inaccuracy of the rainfall estimate, but won't indicate the direction of it. Moreover, high STD may reflect for example a hill or a pit; in the first case it'll probably increase the runoff from the pixel to its neighbours, and in the latter from neighbours to the target pixel. I'm not necessarily suggesting to reject this model term, but use of it is not totally justified, as it won't necessarily indicate any particular tendency per se.**
Agreed. The use of this variable cannot indicate the direction of the total amount of water input at any given location in relation to the rainfall amount over that specific location. We used it as a means to account for regions that experience additional in or out-flow of groundwater, as opposed to being directly rainfed. The use of elevation standard deviation was found to be of significance in our model. However, we agree that its use may not be fully clear in this context, and we will therefore remove it from our revised model implementation and recalculate the statistics. This will not impact the interpretation of our results.

Over the RoC sub-basins there is less local variability in the elevation within the 0.05 x 0.1 degree pixels we use than for over the DRC (figure 5), For the DRC sub-basins, the percentage of palm vs. hardwood swamp is strongly correlated with the variation in elevation within the modelled regions (figure 5c). However, for the sub-basins we ran our final model over (labelled RoC and Mixed), the elevation standard deviation was the variable that contributed the least to our model predictions, with a p-value of 0.035, compared with $p \ll 0.05$ for most other modelled variables (table 1). Removing this variable from our model will not result in a significant impact on the model outputs.

**Row 363: what is a "blackwater river"?**
We agree that this is not currently made clear within the text. We will add the definition to the manuscript. Rivers, or regions along a river, can be described as black-, white- or clear-water, referring to their colour, the speed at which they flow, and their nutrient content. Blackwater rivers are generally slow moving, dark in colour and low in nutrients due to the leaching of acidic tannins from swamp vegetation into the water. Whitewater rivers are faster flowing, with suspended sediment and higher nutritional content.

**Row 419: I'm not sure if "contribute significantly" is the best way to say here; rather, they enable to model the vegetation types at a reasonable accuracy**
Agreed, thanks for this suggestion. We will modify this line as suggested.

---

## Author Comment (AC2)

**Response to reviewer 2**

We would like to thank reviewer 2 for taking the time to review our manuscript and to provide such a detailed response and comments. Two major points have been cited by this reviewer which we address in detail here, along with responding to their additional comments. We are grateful for these comments and are confident that the changes we will make in response will result in an improved manuscript. With specific reference to the second major comment, we will run a modified model implementation, following the reviewer's suggestion. This is straightforward to run, and we do not expect it to change the overall conclusions of the manuscript. We provide here our responses to each comment the reviewer made and how we will modify the manuscript as a result. The reviewer's comments are in bold, and our response in normal text.

**(1) Ground truth data**
**The manuscript uses as ground truth data the mapping product of Crezee et al. 2022. The supplementary Figure 1 of the paper by Crezee et al. shows the nine remote-sensing products that were used to map peat-associated vegetation, i.e. the ground truth data used here in the work of Georgiou et al. Three of the nine input variables were based on elevation data. The fact that detailed elevation data was already used in the generation of the ground truth data conceptually prohibits that in Georgiou et al. elevation data is again used to build a regression model. In Georgiou et al., it is found that peat swamp vegetation is mainly a function of elevation. Knowing that the ground truth data was already created with elevation data makes this a trivial finding. Any discussion in Georgiou et al. on the influence of elevation-based variables is far-fetched given this fundamental problem of the ground truth data. To analyze the influence of elevation, the authors would need to work with ground truth data that is e.g. solely based on optical and microwave satellite signatures, but not on elevation.**

We agree that there is a circular relationship between our use of elevation in the model, and elevation being used in the original mapping (although a different elevation product and at a different resolution). Within the manuscript we describe our implementation of three models, using different sets of input variables: (i) One which includes rainfall, temperature and elevation variables, and which leads to the most accurate predictions ($p<0.05$, $R^2=0.75$ in table 1); (ii) a seasonal-rainfall-only model we tested which didn't include the elevation but leads to significant correlation between the dry season rainfall variables and peat swamp type in rainfed regions ($p<0.05$, $R^2=0.5$); and (iii) a seasonal temperature only model, which made significant predictions but less so than the other models (p-value$<0.05$, $R^2=0.39$). The inclusion of elevation as a model variable does improve the model's accuracy, but the significance of the relationships between seasonal rainfall and temperature variables holds without inclusion of the elevation. We therefore have confidence that our conclusions would still stand even if elevation wasn't included as a model variable.

The elevation's significance is most likely a proxy for additional runoff inputs and outputs as a result of the slope or proximity to river edges that may flood over. We include a more detailed interpretation of the elevation's contribution within sections 5.4 and 5.6.

We used the Crezee et al. (2022) data as the ground truth data as the statistics showed it to have reasonable accuracy when compared with ground-based observations of peat swamp types. We currently mention within the text that:

> It is important to note that the vegetation type map data that this study was based on has some inherent inaccuracy due to it being based on modelled data. Crezee et al. (2022) estimate their peatland distribution model to have a Matthews correlation coefficient of 80%. Misclassification within the underlying mapped data between palm and hardwood swamp pixels, or due to the possibility of mixed vegetation types that do not conform to either of these two main classes, and are therefore misclassified, may be responsible for the outlying predictions.

The Crezee et al. (2022) map used elevation as an input product. However, their mapped product contains five land-type classes, the distinctions between which are better defined using L-band radar data. As such, their modelled significance of the elevation in creating the map was low, and the mapping was largely based on satellite data (ALOS). We believe that our use of elevation is sufficiently independent of their use of it due to the difference in number of land classes used, and also that we're looking at variations over a different spatial scale (the land class pixels at the original mapped resolution of ~50m were aggregated into 0.05 by 0.1 degree pixels for our study). Additionally, we use a different elevation product, the MERIT hydrologically adjusted DEM, available at ~90 m resolution (Figure 2a). It was developed using the NASA SRTM3 and the JAXA AW3D  30 m DEM as baseline products, with bias, noise and tree-height corrections applied to approximate the terrain elevation, and with the Congo Basin being included as one of the focal regions when developing the product (Yamazaki et al., 2017).

**(2) Division into sub-basins and random cross validation**
**The distribution of hardwood trees and palm shows patterns with clear spatial autocorrelation structure. The authors ignored this structure in their 'random' cross-validation approach at sub-basin scale, and thus seriously underestimated predictive error and likely have built overfitted models with non-causal predictors. For details I refer to the highly cited methodological paper of Roberts et al. 2017 on data structure and cross validation (see below). The derived models at sub-basin scale that use, apart from elevation, many different types of climatological-based variables are therefore highly questionable. The authors would need to show that the proposed climatological variables are reliable in a stratified cross-validation that acknowledges the spatial auto-correlation of the data. I believe that this would require an aggregation of sub-basins into larger regions. Perhaps one model for RoC and one for DRC in which one perhaps e.g. stratify the cross-validation by sub-basins (= not building a model for each sub-basin but building a model for four sub-basins and cross-validate against the fifth). Only variables that survive as reliable predictors in such a stratified cross-validation could be used as basis for an interpretation of optimal vegetation conditions**

We agree that there is spatial autocorrelation within the data to some extent. We can mitigate this with some further refinements to our model, as the reviewer suggests. We thank the reviewer for their suggestion. Additionally, this has already been mitigated for and tested to some extent within our current methods:

(i) The original mapped data was at ~50m resolution. The palm and hardwood swamp composition within each of these pixels as a percentage of the total palm and hardwood swamp composition was then calculated over pixels of 0.05 x 0.1 degree resolution, before being used within our model analysis.. These pixels contain varying amounts of other land types (terra firme, savannah and water). The use of these derived pixel groupings, as opposed to the original neighbouring 50m resolution vegetation classification pixels will have mitigated for some of the spatial correlation inherent in the original data.

(ii) We tested the model stability and found it to be good. This involved running 10 different random combinations of train-test split data, within each sub-basin grouping (see table S.4 in the supplementary information linked and copied below), with each 80% train, 20% test split producing very similar statistics.

**Table S.4.** Test of the model stability. Z-score beta coefficients for the mean ($\mu$) model using 10 different random-seed numbers within the Scikit-learn algorithm which divides the data into 80% training and 20% test data sets. We initialised the random-seed value to 0 within our final model.

| Feature | \multicolumn | | | | | | | | | | | |
|---|---|---|---|---|---|---|---|---|---|---|---|---|
| | **0** | **12** | **17** | **36** | **42** | **78** | **112** | **247** | **571** | **1000** | **Mean** | **Std dev** |
| Intercept | **-2.83** | -2.86 | -3.03 | -3.11 | -2.74 | -2.81 | -2.80 | -2.82 | -2.88 | -2.79 | -2.87 | 0.11 |
| elevation | **-4.64** | -4.72 | -4.89 | -4.83 | -4.68 | -4.78 | -4.56 | -4.68 | -4.67 | -4.92 | -4.74 | 0.11 |
| elevation std dev | **-0.30** | -0.24 | -0.34 | -0.46 | -0.15 | -0.13 | -0.37 | -0.27 | -0.29 | -0.01 | -0.26 | 0.13 |
| Dry rainfall diff | **3.23** | 2.95 | 3.21 | 3.17 | 3.21 | 3.16 | 3.18 | 3.21 | 3.24 | 3.10 | 3.17 | 0.09 |
| DJF rainfall | **0.67** | 0.53 | 0.51 | 0.48 | 0.63 | 0.59 | 0.77 | 0.66 | 0.54 | 0.56 | 0.59 | 0.09 |
| DJF Tmax | **-1.28** | -1.20 | -1.36 | -1.35 | -1.29 | -1.25 | -1.21 | -1.36 | -1.38 | -1.24 | -1.29 | 0.07 |
| Total wet season | **0.73** | 0.63 | 0.72 | 0.73 | 0.69 | 0.66 | 0.67 | 0.65 | 0.73 | 0.66 | 0.69 | 0.04 |
| SON Tmax | **-0.35** | -0.45 | -0.40 | -0.40 | -0.30 | -0.34 | -0.35 | -0.41 | -0.32 | -0.38 | -0.37 | 0.04 |
| Statistics | | | | | | | | | | | | |
| $R^2$ | **0.75** | 0.74 | 0.74 | 0.75 | 0.74 | 0.74 | 0.73 | 0.74 | 0.73 | 0.75 | 0.74 | 0.01 |
| log-likelihood | **939.5** | 921.6 | 920.6 | 924.6 | 935.4 | 924.9 | 880.9 | 920.7 | 916.7 | 938.9 | 922.4 | 16.7 |

Supplementary information link:
https://egusphere.copernicus.org/preprints/2022/egusphere-2022-580/egusphere-2022-580-supplement.pdf

Some of the other graphs within the supplementary information also show the significant relationships between the seasonal rainfall variables and the peat swamp type composition (Figures S1 to S5).

It will not be possible to fully eliminate spatial auto-corrrelation, due to the large contiguous regions of peatland. To further ensure that we can minimise it, we will test, and describe within the revised manuscript a model that uses spatial partitioning for the train-test split, rather than the random split we previously used.

Our code is written in Python, and we will test the use of the open source toolbox, *Museo ToolBox,* described in the paper linked below to make spatial cross-validation possible within our use-case. This employs a spatial leave-one-out cross-validation approach.
https://link.springer.com/article/10.1007/s10994-021-05972-1

We used sub-basins as a means to delineate regions with different hydrological characteristics (e.g., some are almost fully rainfed, while others receive river inputs, and some are more affected by run-off). Within the model a constant has been derived for each individual basin, such that all the sub-basin models share the same multiplying values for the parameters, but each with a different constant value (intercept). This makes it more complex to select training and test points from across the sub-basin divide. The sub-basins can be grouped into larger combined ones, but by doing so the model's parameter coefficients will change. Given the strong correlations between the seasonal climatological variables and the swamp type composition, we expect the relationships to be close to the modelled ones currently presented in the manuscript. We agree that there will be spatial correlation within each model, but this may be impossible to avoid completely in any spatial analysis where the region needs to be divided by hydrological characteristics.

More simply than re-running the models over aggregated sub-basins, we could remove the word 'independent' when referring to the test data, and explain that there will be some spatial correlation that is difficult to avoid when working with hydrologically distinct sub-regions.

Detailed comments:

**Line 35:**
**Harmonize use of Pg C and Gt C in the paper.**
Agreed, thanks. We will modify all references to use Pg C.

**Line 98:**
**A useful variable might be the 'topographic wetness index' that combines subbasin area and local slope to estimate ground- and surface water impacts on soil wetness (e.g. Kopecky et al. 2021).**
Thanks for this suggestion. We originally calculated TPI from the 90m MERIT Hydro slope data, however we didn't consider it as a candidate for inclusion in our model as its standard deviation was too large over the 0.05 x 0.1 degree pixel resolution we ran our model at.

**Line 163-164: Sentence unclear**
Agreed that this is not clear. Thanks for bringing this to our attention. The original sentence read: The inclusion of too many multi-collinear variables in the regression method we use also results in non-convergence of the model. It will be modified to say: The beta-regression model is unable to converge on a solution and to provide meaningful output when there is a high degree of multi-collinearity between input variables.

**Line 210:**
**It's not 'train-test' since "test" data needs to be independent. With a random sampling, test data points are spatially auto-correlated with training points, thus they are not independent.**
Agreed. We will test re-running our models using train-test splits in a way that mitigates spatial correlation. If we determine that spatial correlation cannot be fully mitigated for, then we will make it clear that our data split cannot be described as being fully independent. Thanks for your thoughts on this.

**Line 261:**
**Also for RoC sub-basins not all show a positive correlation b/w palm fraction and annual rainfall (Roc5 show negative correlation)**
We found the correlation with annual rainfall to be a weakly significant one and not consistent between sub-basins, and we do not use this variable in the final model selections, although we have included the relationships within the results section as the relationship between swamp type composition and annual rainfall is of higher significance in the DRC sub-basins to the right of the Congo river, and this is useful in our discussion of a potential upper limit on water inputs for palm swamp dominance within a region.

**Figure 6:**
**spatial variation of precipiation in RoC is only 100 mm, ~ 6-7%. In this example, it's quite likely that this trend will prove unreliable in a stratified cross validation.**

We don't use annual precipitation within the final model. The percentage variation in the dry season rainfall variables is much higher for the largely rainfed sub-basins on the left side of the Congo river.

**Line 455:**
**Is there any physiological indication why palms should be less able to tolerate wetness than hardwood trees? Based on the methodological problems of the study, I found the discussion on the optimal water amounts for palms based on the negative correlation of palms with rainfall far-fetched.**

This is an interesting question. We're currently unsure of the physiological indications as to why palms should be less able to tolerate higher levels of wetness. This is still a hypothesis based on the relationships between rainfall input and swamp type composition in the sub-basins to the right of the Congo river, which were not included in the model due to the significant differences in hydrological mechanisms between peatlands located to the the left and right banks of the Congo river. Field observations of the physiological and morphological characteristics specific to Raphia laurentii would help to establish the underlying reasons for the differences in optimal climatological conditions between the palm and hardwood swamp vegetation  types present in the Cuvette Centrale.